# Unique S-scheme heterojunctions in self-assembled $TiO_2$/$CsPbBr_3$ hybrids for $CO_2$ photoreduction

Feiyan Xu [1,2], Kai Meng [1], Bei Cheng [1], Shengyao Wang [3✉], Jingsan Xu [4✉] & Jiaguo Yu [1,2✉]

Exploring photocatalysts to promote $CO_2$ photoreduction into solar fuels is of great significance. We develop $TiO_2$/perovskite ($CsPbBr_3$) S-scheme heterojunctions synthesized by a facile electrostatic-driven self-assembling approach. Density functional theory calculation combined with experimental studies proves the electron transfer from $CsPbBr_3$ quantum dots (QDs) to $TiO_2$, resulting in the construction of internal electric field (IEF) directing from $CsPbBr_3$ to $TiO_2$ upon hybridization. The IEF drives the photoexcited electrons in $TiO_2$ to $CsPbBr_3$ upon light irradiation as revealed by in-situ X-ray photoelectron spectroscopy analysis, suggesting the formation of an S-scheme heterojunction in the $TiO_2$/$CsPbBr_3$ nanohybrids which greatly promotes the separation of electron-hole pairs to foster efficient $CO_2$ photoreduction. The hybrid nanofibers unveil a higher $CO_2$-reduction rate (9.02 μmol $g^{-1}$ $h^{-1}$) comparing with pristine $TiO_2$ nanofibers (4.68 μmol $g^{-1}$ $h^{-1}$). Isotope ($^{13}CO_2$) tracer results confirm that the reduction products originate from $CO_2$ source.

[1] State Key Laboratory of Advanced Technology for Materials Synthesis and Processing, Wuhan University of Technology, Wuhan 430070, P.R. China. [2] Foshan Xianhu Laboratory of the Advanced Energy Science and Technology Guangdong Laboratory, Xianhu Hydrogen Valley, Foshan 528200, P.R. China. [3] College of Science, Huazhong Agricultural University, Wuhan 430070, P.R. China. [4] School of Chemistry and Physics & Centre for Materials Science, Queensland University of Technology, Brisbane, QLD 4001, Australia. ✉email: wangshengyao@mail.hzau.edu.cn; jingsan.xu@qut.edu.au; jiaguoyu93@whut.edu.cn

The depletion of fossil fuels and continuous $CO_2$ emissions have caused emerging global energy and environmental crises[1–5]. The photoreduction of $CO_2$ into renewable fuels with solar energy is recognized as a potential solution to solve above issues[6–10]. As a chemically inert, nontoxic and earth-abundant photocatalyst, $TiO_2$ is supposed to be proverbially utilized for $CO_2$ photoreduction[11–13]. However, like the majority of unitary photocatalysts, the photocatalytic efficiency of $TiO_2$ is still far away from the practical requirements largely due to its rapid electron–hole recombination[14,15]. Hybridizing $TiO_2$ with another semiconductor with a suitable band structure is a widely adopted strategy to tackle this issue owing to the efficient separation of photoinduced electron–hole pairs[16–20]. Therefore, it is of significance to explore or design a $TiO_2$-based heterojunction to improve the photocatalytic $CO_2$ reduction performance.

$CsPbBr_3$, a typical material of halide perovskites, has attracted significant scientific interest in optoelectronic applications owing to its outstanding properties, including narrow photoemission, high photoluminescence quantum yield, tunable bandgap, and competing optoelectronic properties[21–24]. Inspired from the achievements in optoelectronic applications, $CsPbBr_3$ is a potential candidate for conducting efficient photocatalysis[25,26]. $CsPbBr_3$ quantum dots (QDs) have recently been hybridized with 2D graphene oxide[27] and porous $g$-$C_3N_4$[28] for $CO_2$ photoreduction. Nevertheless, in these cases, the electrons in the conduction band of $CsPbBr_3$ transferred into graphene and $g$-$C_3N_4$, forming Schottky and type-II heterojunctions, respectively, sacrificing the reduction ability of the photoinduced electrons despite achieving better charge separation. Very recently, an S-scheme heterojunction composed of two n-type semiconductors has been proposed[29,30]. The transfer path of photogenerated charge carriers at interfaces is like an "S" figure, enabling the heterojunctions to have the highest redox ability. The S-type charge transportation correlates with the band bending and internal electric field (IEF) at the junction. The n-type nature and

remarkably different work functions of $TiO_2$ and $CsPbBr_3$ suggest a high possibility of forming S-scheme $TiO_2$/$CsPbBr_3$ heterojunctions. Up to now, however, constructing perovskite $CsPbBr_3$ with $TiO_2$, an emerging photoactive material and the most widespread photocatalyst, for efficient $CO_2$ photoreduction has not yet been reported.

Herein, we report on a unique $TiO_2$/$CsPbBr_3$ S-scheme heterojunction built by electrostatic self-assembly of $TiO_2$ nanofibers and $CsPbBr_3$ QDs for boosted photocatalytic $CO_2$ reduction. $TiO_2$ nanofibers show no aggregation upon dispersion in solution and thereby retain their phototactically active sites exposed on the surface. Meanwhile, randomly stacked $TiO_2$ nanofibres readily form a loose network, facilitating the adsorption–desorption and transportation of reactants and products. More importantly, the $TiO_2$ nanofibres are composed of small nanocrystals, possessing interparticle voids and rough surface, which make $TiO_2$ nanofibres an ideal host to anchor $CsPbBr_3$ QDs. Experimental study and density functional theory (DFT) calculation verify the presence of IEF in the unique $TiO_2$/$CsPbBr_3$ heterojunction, which separate photoinduced charge carriers more efficiently. We argue the formation of the S-scheme charge transfer route at $TiO_2$/$CsPbBr_3$ interfaces upon light irradiation. The obtained $TiO_2$/$CsPbBr_3$ heterojunction shows a superior activity for reducing $CO_2$ into solar fuels under UV–visible-light irradiation. This work provides a point of view in $TiO_2$-based photocatalyst for efficient $CO_2$ photoreduction driven by the S-scheme electron transfer route.

## Results and discussion

**Characterization of as-prepared $CsPbBr_3$ QDs.** Transmission electron microscopy (TEM) images with different magnifications are shown in Fig. 1a, b. The $CsPbBr_3$ QDs were of nanocubes with a size of 6–9 nm (inset in Fig. 1a). High-resolution TEM (HRTEM) image (Fig. 1c) showed lattice spacings of 0.413 nm, corresponding to the (110) facets of $CsPbBr_3$. As-prepared

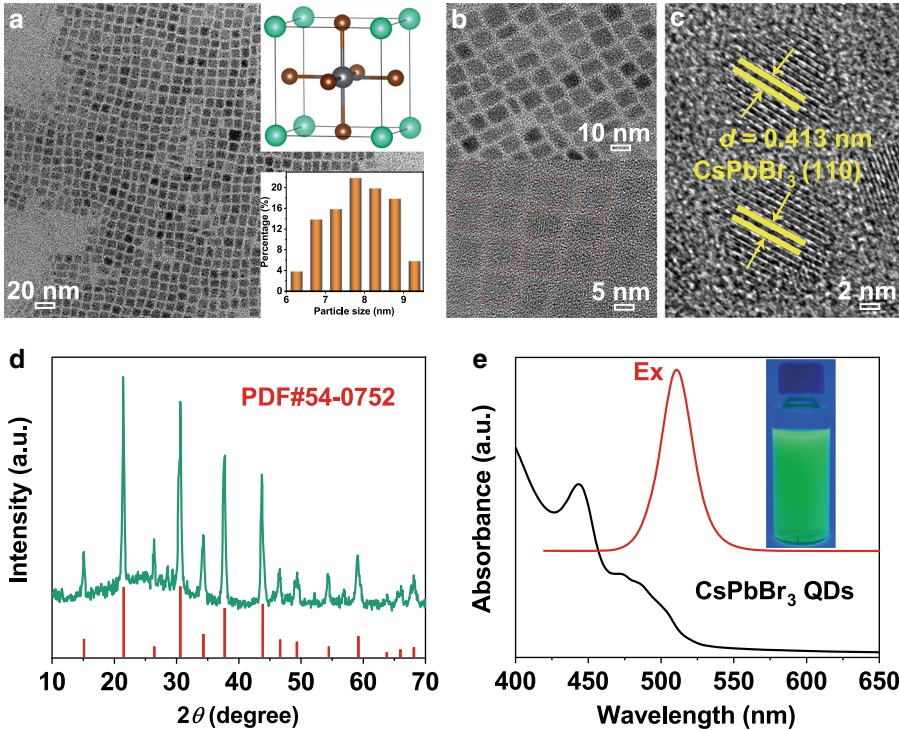

**Fig. 1 Characterization of CsPbBr₃ QDs. a, b** Transmission electron microscopy (TEM) image and corresponding size distribution (lower right inset of panel **a**), the geometrical structure (upper right inset of panel **a**), **c** high-resolution TEM (HRTEM) image, **d** X-ray diffraction (XRD) pattern, and **e** UV–vis absorption (black line) and PL emission (red line). Inset shows the photograph of CsPbBr₃ QDs colloidal solutions in hexane under UV light of 365 nm.

$CsPbBr_3$ QDs were of cubic phase (JCPDS No. 54-0752) as revealed by X-ray diffraction (XRD) pattern (Fig. 1d). The UV–vis absorption spectrum of $CsPbBr_3$ QDs revealed strong bands at 450 and 500 nm (Fig. 1e). The corresponding photoluminescence (PL) spectrum unfolded a narrow emission at 520 nm, agreeing with previous reports[21,31]. Accordingly, the QDs solution showed a bright green fluorescence under 365 nm UV light (inset of Fig. 1e).

**Characterization of $TiO_2/CsPbBr_3$ heterojunction**. The $TiO_2/CsPbBr_3$ heterojunction was synthesized via electrostatic self-assembly of $TiO_2$ nanofibers and $CsPbBr_3$ QDs. Moreover, the minimization of the surface energy of the QDs should also be responsible for their adsorption to the $TiO_2$ nanofibers. The $TiO_2/CsPbBr_3$ hybrids were denoted as TCx, where T and C denote $TiO_2$ and $CsPbBr_3$ QDs, respectively; x represents the weight percentage of $CsPbBr_3$ with respect to $TiO_2$. The phase structures of $TiO_2$, TC2, and TC4 were determined via XRD analysis (Supplementary Fig. 1). $TiO_2$ nanofibers showed intensive reflections belonging to anatase (JCPDS No. 21-1272) and rutile (JCPDS No. 21-1276) phases. TC2 showed a similar XRD pattern with pristine $TiO_2$, where the reflections of $CsPbBr_3$ QDs cannot be distinguished due to their low content. Apart from the characteristic reflections of $TiO_2$, TC4 showed additional reflections at 21.5° and 30.6°, which corresponded to the (110) and (200) planes of $CsPbBr_3$ QDs, confirming the formation of $TiO_2/CsPbBr_3$ nanohybrids. The morphology and crystalline phase of pristine $TiO_2$ (Supplementary Fig. 2a) exhibited a porous nanofibrous shape with an average diameter of 200 nm. The porous feature was further revealed by the $N_2$ sorption isotherms of TCx (Supplementary Fig. 3). All the TCx samples showed similar pore size distributions with a wide range of 10–20 nm, much larger than the size of $CsPbBr_3$ QDs (6–9 nm). The resultant specific surface areas ($S_{BET}$), pore volumes ($V_p$), and average pore sizes ($d_p$) presented a volcano shape with increasing the loading of $CsPbBr_3$ QDs (Supplementary Table 1). At a low QDs loading (<2 wt.%), TCx showed an increased $S_{BET}$ and reached the maximum value at TC2 because the low filling enables QDs to deposit onto the inner wall of $TiO_2$ mesopores. Such island-like QDs on the inner wall contribute additional specific surface area for the hybrid. When the QDs loading was further increased, QDs would aggregate in $TiO_2$ mesopores and the island-like distribution vanished, which thereby resulted in a decrease of $S_{BET}$. The HRTEM image (Supplementary Fig. 2b) showed clear lattice spacings of 0.352 and 0.325 nm, corresponding to anatase (101) and rutile (110) d-spacings, respectively. After the assembling process, the QDs were uniformly deposited on the $TiO_2$ nanofibers (Fig. 2a, b). The lattice spacings of anatase and rutile phase $TiO_2$, as well as $CsPbBr_3$ QDs, appeared in the HRTEM image, as shown in Fig. 2c, confirming the formation of $TiO_2/CsPbBr_3$ nanohybrids. The energy-dispersive X-ray spectroscopy (EDX) spectrum of TC2 (Fig. 2d) revealed the existence of Cs, Pb, and Br apart from the dominant Ti and O elements. All the elemental mappings overlapped perfectly (Fig. 2e). Fourier-transform infrared (FTIR) spectra showed the presence of (Ti)–OH on $TiO_2$ and organic residues on QDs (Supplementary Fig. 4a, b)[32]. The (Ti)–OH signal weakened upon QDs deposition owing to the shielding effect of QDs. All the results confirmed the successful electrostatic assembly of $TiO_2$ nanofibers and $CsPbBr_3$ QDs.

The optical absorption of the samples was investigated by UV–vis diffuse reflectance spectrometer (DRS) (Supplementary Fig. 5a). The absorption edges of pristine $TiO_2$ nanofibers and $CsPbBr_3$ QDs were located at 400 and 550 nm, corresponding to the bandgap energy of 3.10 and 2.24 eV, respectively (Supplementary Fig. 5b). In comparison with pristine $TiO_2$, TCx showed

two obvious absorption edges belonging to $TiO_2$ and $CsPbBr_3$ QDs, and exhibited slightly enhanced UV and visible-light harvesting when increasing the amount of $CsPbBr_3$ QDs owing to the strong light-harvesting capability of perovskite QDs. Note that the calculated bandgap energy of $TiO_2$ and $CsPbBr_3$ in TC4 was different from their intrinsic bandgap, implying that there exist electrostatic attraction and interaction between $TiO_2$ and $CsPbBr_3$ during the hybridization.

X-ray photoelectron spectroscopy (XPS) was further performed to explore the chemical states of the resultant samples. The survey XPS spectrum (Supplementary Fig. 6a) showed the presence of Cs, Pb, and Br elements within TC2, as well as Ti and O. The ex-situ Ti 2p XPS spectra of $TiO_2$ and TC2 (Fig. 3a) showed symmetrical Ti 2p doublets of $Ti^{4+}$ ions. The O 1s XPS spectra (Fig. 3b) revealed the presence of lattice oxygen (529.3 eV) and –OH surface group (531.2 eV). Interestingly, TC2 showed a weaker XPS signal of –OH than pristine $TiO_2$, which was also attributed to an increase of QDs over $TiO_2$ nanofiber surface and was in agreement with the above FTIR results. The Br 3d-binding energies (BEs) of $CsPbBr_3$ QDs were 67.8 and 69.8 eV, corresponding to Br $3d_{5/2}$ and Br $3d_{3/2}$, respectively (Fig. 3c). Noticeably, the BEs of Ti 2p and O 1s in TC2 were shifted by 0.2 eV toward a lower BE in comparison with those of pristine $TiO_2$, while the Cs 3d, Pb 4f (Supplementary Fig. 6c, d) and Br 3d BEs of TC2 became more positive as compared with those of QDs, indicating that the electrons transferred from $CsPbBr_3$ QDs to $TiO_2$ upon hybridization due to the difference of their work functions. Such electron transfer created an IEF at interfaces pointing from QDs to $TiO_2$, facilitating the construction of S-scheme $TiO_2/CsPbBr_3$ heterojunction without any redox mediator, which would efficiently separate the charge carriers and thus promote the $CO_2$ photoreduction[33–35].

Work function ($\Phi$), as another important parameter to study the electron transfer within duplicate semiconductor heterostructures, can be estimated from the energy difference of vacuum and Fermi levels according to the electrostatic potential of a material. As shown in Fig. 3d–f, the work function of anatase $TiO_2$ (101), rutile $TiO_2$ (110), and $CsPbBr_3$ QDs (001) were 7.18, 7.08, and 5.79 eV, respectively, indicating that both anatase and rutile $TiO_2$ have lower Fermi levels than $CsPbBr_3$ QDs. When they contacted with each other, electrons would flow from $CsPbBr_3$ to anatase and/or rutile $TiO_2$ to enable the phases at the same Fermi level and definitely created an IEF at $TiO_2/CsPbBr_3$ interfaces. These results were absolutely consistent with above ex-situ XPS results and beneficial to the charge separation and $CO_2$ photoreduction activity.

**$CO_2$ photoreduction activity of $TiO_2/CsPbBr_3$ hybrids**. The $CO_2$ photoreduction activity of resultant samples was measured in a closed gas-circulation system (Supplementary Fig. 7) with a Quartz and Pyrex glass hybrid reaction cell (Supplementary Fig. 8) and the photocatalytic reduction products consisted of a majority of CO and a small amount of $H_2$. The original chromatograms for the reduction of $CO_2$ on sample TC2 are shown in Supplementary Fig. 9. Control experiments (Supplementary Fig. 10 and Table 2) showed that neither $H_2$ nor CO was detected in the dark or in the absence of $CO_2$, suggesting that the light irradiation and input $CO_2$ were indispensable for the photocatalytic reaction. As shown in Fig. 4a, b, pristine $TiO_2$ and $CsPbBr_3$ QDs exhibited relatively lower production rates of $H_2$ (0.12 and 0.06 $\mu mol\ g^{-1}\ h^{-1}$, respectively) and CO (4.68 and 4.94 $\mu mol\ g^{-1}\ h^{-1}$, respectively), resulting from the rapid charge recombination. Note that the $H_2$ and CO productions were greatly enhanced with increased loading of QDs, and the generation of CO reached a maximum rate (9.02 $\mu mol\ g^{-1}\ h^{-1}$) with a

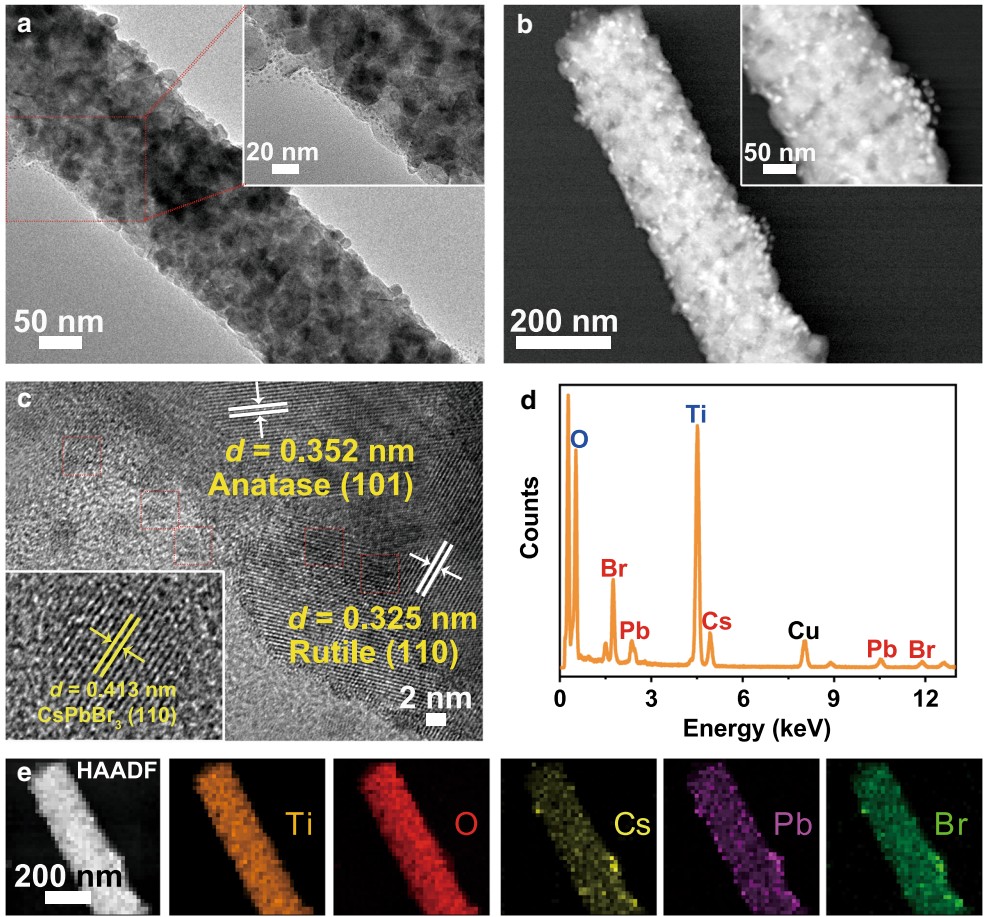

**Fig. 2 Morphology and structure of TiO₂/CsPbBr₃ heterojunction. a–c** Transmission electron microscopy (TEM), STEM, and high-resolution TEM (HRTEM) images of TC2, **d** EDX spectrum of TC2, and **e** high-angle annular dark-field (HAADF) image and EDX elemental mappings of Ti, O, Cs, Pb, and Br elements in TC2.

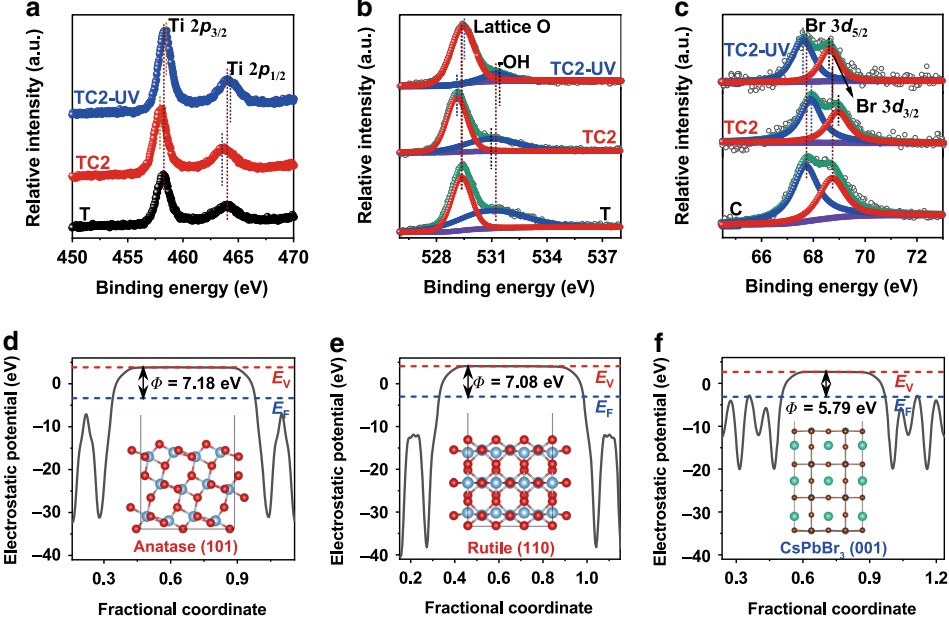

**Fig. 3 Electron transfer between TiO₂ and CsPbBr₃ quantum dots (QDs).** In-situ and ex-situ X-ray photoelectron spectroscopy (XPS) spectra of **a** Ti 2p, **b** O 1s, and **c** Br 3d of TiO₂, CsPbBr₃, and TC2. In-situ XPS spectra were collected under UV–vis light irradiation. The electrostatic potentials of **d** anatase TiO₂ (101), **e** rutile TiO₂ (110), and **f** CsPbBr₃ (001) facets. The blue, red, green, gray, and brown spheres stand for Ti, O, Cs, Pb, and Br atoms, respectively. Blue and red dashed lines indicate the Fermi and vacuum energy levels.

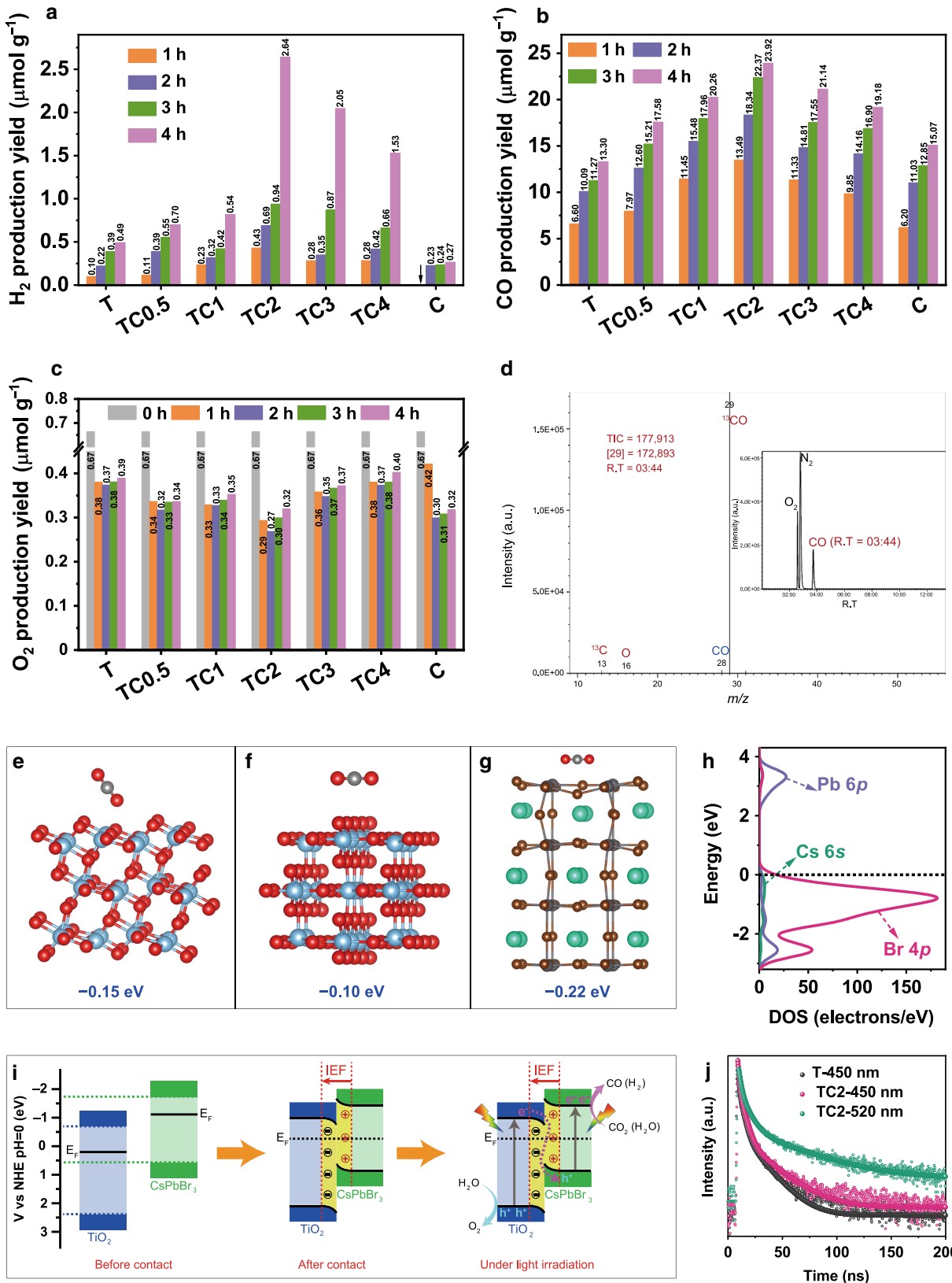

relatively high selectivity (95%) over TC2, due to the efficient charge separation of $TiO_2/CsPbBr_3$ heterostructure. Further increasing $CsPbBr_3$ QDs amount would be detrimental to the photocatalytic activity (e.g., TC3 and TC4), because the overloading of $CsPbBr_3$ could shield the light absorption of $TiO_2$ and decrease $S_{BET}$ of the nanohybrids. Interestingly, with the reaction

time went on, the amount of $O_2$ decreased first and then increased, as shown in Fig. 4c. The initial $O_2$ in the system came from the input high-purity $CO_2$. In the first two hours of photocatalytic $CO_2$ reduction, the fresh materials exhibit relatively strong reactivity of photoreduction. As a competitive reaction to $CO_2$ reduction, the consumption rate of $O_2$ ($O_2 + e^- \rightarrow \cdot O_2^-$) was

**Fig. 4 CO₂ photoreduction performance and the photocatalytic mechanism of S-scheme heterojunction.** Photocatalytic activities of $CO_2$ reduction over $TiO_2$, TC$x$, and $CsPbBr_3$ quantum dots (QDs) during 4-h experiment performed under UV–vis light irradiation: time course of **a** $H_2$, **b** CO, and **c** $O_2$ production yields. The initial $O_2$ concentrations were normalized. **d** Mass spectra of $^{13}CO$ and total ion chromatography (inset) over TC2 in the photocatalytic reduction of $^{13}CO_2$. Optimized structures of $CO_2$ molecule adsorbed on **e** anatase $TiO_2$ (101), **f** rutile $TiO_2$ (110), and **g** $CsPbBr_3$ (001) facets. The blue, red, green, gray, and brown spheres stand for Ti, O, Cs, Pb, and Br atoms, respectively. **h** The DOS of $CsPbBr_3$. **i** Schematic illustration of $TiO_2$/$CsPbBr_3$ heterojunction: internal electric field (IEF)-induced charge transfer, separation, and the formation of S-scheme heterojunction under UV–visible-light irradiation for $CO_2$ photoreduction. **j** Time-resolved photoluminescence (TRPL) spectra of $TiO_2$ (T) and TC2 at emission wavelengths of 450 and 520 nm, respectively.

much higher than the production rate at the initial 2 h, while in the following 2 h, the production rate of $O_2$ was higher than the consumption rate, the total amount of oxygen and the ratio of oxygen:nitrogen have increased to a certain extent.

The recyclability and stability of TC2 for $CO_2$ photoreduction were investigated (Supplementary Fig. 11). After four times cycles, the decay of photocatalytic production yields of $H_2$ and CO were hardly perceptible. To evaluate the photostability of the nanohybrids, we have characterized the recycled photocatalyst using XRD, TEM, XPS, and FTIR. As shown in the XRD pattern (Supplementary Fig. 12a), the used photocatalyst showed no detectable phase change. The TEM image confirms that the QDs did not show obvious aggregation after cycled photocatalytic reactions, and the morphology was well maintained (Supplementary Fig. 12b). The chemical states of the used photocatalyst were also consistent with those of the fresh one, as examined by XPS (Supplementary Fig. 13). The FTIR spectra of TC2 before and after reaction were presented in Supplementary Fig. 14. The characteristic absorbance bands of the aliphatic species from QDs showed no obvious variation, implying that the capping agent of QDs was stable and was not decomposed during the photocatalytic $CO_2$ reduction.

To determine the origin of $CO_2$ photoreduction products, we performed an isotope-labeled carbon dioxide ($^{13}CO_2$) photocatalytic reduction over TC2. Since the amount of products without photosensitizer and hole sacrificial agent was beyond the detection limit of mass spectrometry detector, we added tris(2,2′-bipyridyl)ruthenium(II) chloride hexahydrate ([Ru$^{II}$(bpy)$_3$]Cl$_2$·6H$_2$O)[36] and 1,3-dimethyl-2-phenyl-2,3-dihydro-1H-benzo[d]imidazole (BIH)[37] into the system to promote the photocatalytic activity, which behaved as the photosensitizer and hole sacrificial agent, respectively. In this case, the production yields of $H_2$ and CO were significantly enhanced (Supplementary Fig. 15 and Table 2) and readily detected by gas chromatography–mass spectrometer (GC-MS). As shown in Fig. 4d, the total ion chromatographic peak ~3.44 min corresponded to CO, which produced three signals in the mass spectra. The main MS signal at $m/z = 29$ belonged to $^{13}CO$ and the others ($^{13}C$ at $m/z = 13$ and O at $m/z = 16$) corresponded to the fragments of $^{13}CO$, confirming that the CO product exactly originated from the $CO_2$ photoreduction over $TiO_2$/$CsPbBr_3$[38,39]. In addition, the total ion chromatographic peaks ~2.36 and 2.48 min can be assigned to the $O_2$ and $N_2$, respectively (Supplementary Fig. 16).

The $CO_2$ adsorption of a photocatalyst is an essential step for $CO_2$ photoreduction[40]. Figure 4e–g compared the optimized models of one $CO_2$ molecule adsorbed on anatase $TiO_2$ (101), rutile $TiO_2$ (110), and $CsPbBr_3$ (001) surfaces. Clearly, the adsorption energy ($E_{ads}$) of $CO_2$ onto $CsPbBr_3$ (−0.22 eV) was more negative than that onto anatase and rutile $TiO_2$ (−0.15 and −0.10 eV), which suggests that $CO_2$ molecules adsorbed on $CsPbBr_3$ is more stable than on $TiO_2$. The results also indicate that $CsPbBr_3$ QDs were in favor of the adsorption of $CO_2$ molecules and the photocatalytic $CO_2$ reduction.

To further explore the photocatalytic mechanism, the band structures of $TiO_2$ and $CsPbBr_3$ QDs were investigated. The valence band (VB) potential was obtained by analyzing the VB XPS spectra. As shown in Supplementary Fig. 17a, b, the energy level of valence band maximum (VBM) of $TiO_2$ and $CsPbBr_3$ is 2.39 and 1.03 eV, respectively. Mott–Schottky (M–S) curves showed that $TiO_2$ and $CsPbBr_3$ were of n-type semiconductors and had flat-band potentials of 0.01 eV and −0.51 eV (vs. NHE), respectively (Supplementary Fig. 17c, d). Thus, the band structures of $TiO_2$ and $CsPbBr_3$ QDs can be derived, and the positions of VBM and conduction band minimum (CBM) of $TiO_2$ and $CsPbBr_3$ are shown in Supplementary Fig. 17f.

**Photocatalytic mechanism of S-scheme heterojunction.** From the above analysis, the superior photoreduction activity was ascribed to the stronger $CO_2$ adsorption of $CsPbBr_3$ QDs and the formation of S-scheme heterojunction between $TiO_2$ and $CsPbBr_3$ QDs. As revealed by the above ex-situ XPS and DFT analyses, $TiO_2$ has a lower Fermi level than $CsPbBr_3$ QDs before they contact. Upon hybridization, the electrons preferred to flow from $CsPbBr_3$ QDs to $TiO_2$, which created an IEF at $TiO_2$/$CsPbBr_3$ interfaces pointing from $CsPbBr_3$ to $TiO_2$ and bent the energy bands of $TiO_2$ and $CsPbBr_3$. Upon photoexcitation, the VB electrons of $TiO_2$ and $CsPbBr_3$ jumped to their CBs. Driven by the interfacial IEF and bent bands, the photogenerated electrons in $TiO_2$ CB spontaneously slid toward $CsPbBr_3$ and recombined with the holes in $CsPbBr_3$ VB. The electron-rich $CsPbBr_3$ QDs then acted as active sites and donated electrons to activated $CO_2$ molecules for producing $H_2$ and CO. Noted that Pb was the active site for $CO_2$ photoreduction since the CB of $CsPbBr_3$ was mainly consisted of Pb 6$p$ orbitals as evidenced by the density of states (DOS) of $CsPbBr_3$ (Fig. 4h). Clearly, the transportation of photoinduced charge carriers follows a slide-like pathway, which implies the presence of S-scheme heterojunction between $TiO_2$ and $CsPbBr_3$ QDs. This unique S-scheme charge transfer efficiently separated the photoinduced electron–hole pairs and meanwhile remained the high redox ability of electrons in $CsPbBr_3$ CB and holes in $TiO_2$ VB, respectively. The S-scheme heterostructure of $TiO_2$/$CsPbBr_3$ QDs along with the charge transfer and separation is illustrated in Fig. 4i. Such an S-scheme charge transfer route was strongly evidenced by the in-situ XPS spectra measured under light irradiation. As revealed in Fig. 3a–c and Supplementary Fig. 6c, d, the BEs of Ti 2$p$ and O 1$s$ for TC2 under light irradiation shifted positively by 0.3 eV with reference to those in the corresponding ex-situ spectra. Conversely, the BEs of Cs 3$d$, Pb 4$f$, and Br 3$d$ of TC2 shifted negatively by 0.5 eV. The BE shifts unequivocally proved that the photoexcited electrons in $TiO_2$ CB transferred to $CsPbBr_3$ QDs VB under light irradiation, following an S-scheme pathway, which supported the proposed photocatalytic mechanism.

It is worth mentioning that the $TiO_2$ we used consisted of both anatase and rutile phases, and the charge transfer between the two phases may take place as a result of forming homojunction. As

evidenced by DFT results (Fig. 3d, e), the work function of anatase $TiO_2$ (101) was larger than that of rutile $TiO_2$ (110), indicating that electrons would flow from rutile to anatase and created an IEF at anatase/rutile $TiO_2$ interfaces. Driven by the interfacial IEF, the photogenerated electrons in anatase $TiO_2$ CB spontaneously slid toward rutile $TiO_2$ VB and recombined with the holes in the rutile $TiO_2$ VB. Such transportation of photoinduced charge carriers follows an S-like pathway (S-scheme homojunction) between anatase and rutile $TiO_2$ (Supplementary Fig. 18), which is consistent with our previous work[41]. When $CsPbBr_3$ QDs deposited on $TiO_2$ nanofibers, all possible schematic illustrations between anatase $TiO_2$, rutile $TiO_2$, and $CsPbBr_3$ QDs are shown in Supplementary Fig. 19.

To further prove the efficient charge separation of $TiO_2$/$CsPbBr_3$ S-scheme heterojunction, photoluminescence (PL) emission spectra of the samples were collected (Supplementary Fig. 20). TC2 and TC4 showed a marginally lower PL intensity than $TiO_2$, implying that the presence of $CsPbBr_3$ QDs efficiently retarded the electron–hole recombination in $TiO_2$. To gain a deeper insight into the charge transfer dynamics, the time-resolved photoluminescence (TRPL) spectra of $TiO_2$ and TC2 were recorded at emission wavelengths ($E_W$) of 450 nm and 520 nm (Fig. 4j), corresponding to the maximum fluorescence emissions of $TiO_2$ and QDs, respectively. The fitted decay curves disclose the lifetime ($\tau$) and percentage ($Rel.$%) of charge carriers (Supplementary Table 3). The short lifetime ($\tau_1$) corresponds to radiative recombination of the carriers (denoted as $\tau_1$-carriers), while the long lifetimes ($\tau_2$ and $\tau_3$) correspond to non-radiative recombination and energy-transfer process[42]. Note that the un-recombined $\tau_1$-carriers will participate in surface photocatalytic reaction. Thus, the decrease of $\tau_1$-carrier percentage implies radiative recombination inhibited. At $E_W = 450$ nm, only $TiO_2$ showed a fluorescence emission signal. As shown in Supplementary Table 3, TC2 had a lower percentage (36.27%, 450 nm) of $\tau_1$-carriers than pristine $TiO_2$ (37.98%, 450 nm), suggesting the radiative recombination over $TiO_2$ was inhibited upon QDs deposition due to the formation of S-scheme heterojunction[43,44]. Further, a similar decrease in $\tau_1$-carrier percentage was also observed at $E_W = 520$ nm. Notably, TC2 showed longer lifetime than pristine $TiO_2$ due to the transfer of the electrons in $TiO_2$ CB to QDs VB. Therefore, it is not surprising that the TC2 composite sample exhibited enhanced photocatalytic $CO_2$ reduction performance.

The electrochemical impedance spectra (EIS) (Supplementary Fig. 21a) showed the samples with $CsPbBr_3$ QDs exhibited smaller semicircle compared to pure $TiO_2$ and revealed lower charge-transfer resistance. The polarization curves of $TiO_2$ and TC2 under light irradiation (Supplementary Fig. 21b) showed that the overpotential for TC2 was much lower than that of $TiO_2$, indicating that $TiO_2$/$CsPbBr_3$ hybrids presented better reduction capability than that of $TiO_2$. These results proved that $CsPbBr_3$ QDs, as an emerging semiconductor, could form S-scheme heterojunction with $TiO_2$ to promote the electron transfer and separate the electron–hole pairs for efficient $CO_2$ photoreduction.

In summary, an S-scheme $TiO_2$/$CsPbBr_3$ heterojunction synthesizes through an electrostatic assembly method. The resulting $TiO_2$/$CsPbBr_3$ heterojunction reveals an enhanced activity toward $CO_2$ photoreduction under UV–visible-light irradiation due to the IEF-induced, more efficient charge separation between $TiO_2$ and $CsPbBr_3$. DFT calculations reveal the work function of $TiO_2$ was greater than that of $CsPbBr_3$, implying electrons transfer from $CsPbBr_3$ to $TiO_2$ upon hybridization and thus created an IEF at interfaces. The IEF drives photoinduced electrons in $TiO_2$ CB to immigrate to $CsPbBr_3$ VB as evidenced by in-situ XPS analysis, confirming an S-path of charge transfer. Isotope ($^{13}CO_2$) tracer results confirm that the reduction products originate from $CO_2$ source, instead of any contaminant carbon species. This work provides a point of view in the design of photocatalysts with distinct heterojunctions for efficient photocatalytic $CO_2$ reduction.

## Methods

**Synthesis of electrospun $TiO_2$ nanofibers.** All the chemicals were of analytic grade and purchased from Shanghai Chemical Company. Typically, tetrabutyl titanate (TBT, 2.0 g) and poly(vinyl pyrrolidone) (PVP, 0.75 g, MW = 1,300,000) were mixed with ethanol (10.0 g) and acetic acid (2.0 g) to form a transparent pale-yellow solution after magnetic stirring for 5 h. Afterward, the solution was transferred into a 10-mL syringe in an electrospinning setup with a voltage of 20 kV and a solution feeding rate of 2.5 mL h$^{-1}$. The needle-to-collector distance was 10 cm. The collected $TiO_2$ precursor was annealed at 550 °C for 2 h with a heating rate of 2 °C min$^{-1}$ in air.

**Synthesis of perovskite $CsPbBr_3$ QDs.** Briefly, 130 mg of $Cs_2CO_3$ (0.4 mmol) were mixed with octadecylene (ODE, 6 mL) and oleic acid (OA, 0.5 mL) under stirring in a three-neck flask (25 mL). The mixture was dried at 120 °C for 1 h under vacuum and heated to 150 °C under $N_2$ gas to form Cs(oleate) solution, which was stored at room temperature and preheated to 140 °C prior to use. Then 72 mg of $PbBr_2$ (0.196 mmol) was mixed with ODE (5.0 mL), oleylamine (0.5 mL), and OA (0.5 mL) in another flask (25 mL), and was dried under vacuum at 105 °C for 0.5 h. The mixture was heated to 170 °C, and Cs(oleate) (0.45 mL) was rapidly injected under vigorously stirring for 5 s. The reaction was quenched by immersing the flask into an ice-water bath. The obtained product was mixed with 3 mL of hexane and centrifuged at 1208 × g for 2 min to remove aggregates and large particles. The supernatant was precipitated with acetone and centrifuged at 3355 × g for 5 min. As-collected $CsPbBr_3$ QDs were re-dispersed in hexane for further use.

**Preparation of $TiO_2$/$CsPbBr_3$ heterostructures.** Typically, 200 mg of $TiO_2$ nanofibers were dispersed into 20 mL of hexane. A certain amount of $CsPbBr_3$ QDs solution was added into $TiO_2$ suspension under vigorous stirring for 2 h. $TiO_2$ and $CsPbBr_3$ QDs were assembled by electrostatic self-assembly. The mixture was then vacuum-dried at 50 °C for 2 h to form $TiO_2$/$CsPbBr_3$ heterostructures. The products are labeled as TC$x$, where T and C denote $TiO_2$ and $CsPbBr_3$ QDs, respectively; $x$ is the mass percentage of $CsPbBr_3$ QDs.

**Characterization.** XRD was performed on a D/Max-RB X-ray diffractometer (Rigaku, Japan) with Cu Kα radiation. TEM images were observed on a Titan G2 60-300 electron microscope equipped with an EDX spectrometer. UV–visible DRS was collected on a Shimadzu UV-2600 UV–visible spectrophotometer (Japan). XPS was performed on a Thermo ESCALAB 250Xi instrument with Al K$_\alpha$ X-ray radiation. In-situ XPS was conducted under the same condition, except that UV–visible-light irradiation was introduced. FTIR spectra were recorded with an attenuated total reflectance (ATR) mode on Nicolet iS 50 (Thermo Fisher, USA). The PL emission spectra were collected on a fluorescence spectrophotometer (F-7000, Hitachi, Japan). TRPL spectra were recorded on a fluorescence lifetime spectrophotometer (FLS 1000, Edinburgh, UK) at an excitation wavelength of 325 nm. Electrochemical measurements were conducted on an electrochemical analyzer (CHI660C, CH Instruments, Shanghai). Pt wire, Ag/AgCl (saturated KCl), and 0.5 M $Na_2SO_4$ solution functioned as the counter electrode, reference electrode, and electrolyte, respectively. For the working electrode, 20 mg of TC$x$ was ground in 1.0 mL of ethanol and 10 μL of Nafion solution to make a slurry, which was coated onto F-doped $SnO_2$-coated (FTO) glass with an exposed area of 1 cm$^2$. The FTO electrode was then vacuum-dried at 60 °C for 1 h.

**Photocatalytic $CO_2$ reduction.** The photocatalytic $CO_2$ reduction was performed in a gas-closed system equipped with a gas-circulated pump. The apparatus of the system is shown in Supplementary Fig. 7. Typically, 10 mg of photocatalysts, 30 mL of acetonitrile, and 100 μL of water were added in a Quartz and Pyrex glass hybrid reaction cell (Supplementary Fig. 8). The airtight system was completely evacuated by using a vacuum pump. Then ~80 kPa of high-purity $CO_2$ (99.999%) gas was injected. After adsorption equilibrium, the photocatalytic cell was irradiated with a 300 W Xe arc lamp (PLS-SXE300D, Beijing Perfectlight, China), and the reaction system was kept at 10 °C as controlled by cooling water. The $CO_2$-reduction products were analyzed on a gas chromatograph (GC-2030, Shimadzu Corp., Japan) equipped with a barrier discharge ionization detector (BID) and a capillary column (Carboxen 1010 PLOT Capillary, 60 m × 0.53 mm). The column was maintained at 35 °C for 15 min. It was then heated to 180 °C at 20 °C min$^{-1}$, and maintained for another 5 min. Helium was the carrier gas with pressure set to 70 kPa. The temperatures of the injector and BID were set to be 150 and 280 °C, respectively. For comparison, 2 mM of tris(2,2′-bipyridyl)ruthenium(II) chloride hexahydrate ([Ru$^{II}$(bpy)$_3$]Cl$_2$·6H$_2$O) and 10 mM of 1,3-dimethyl-2-phenyl-2,3-dihydro-1H-benzo[d]imidazole (BIH) were added into the photocatalytic system (other parameters were unchanged), which behaved as the photosensitizer and hole

sacrificial agent, respectively. A series of control experiments were also conducted, and the results are summarized in Supplementary Table 2.

**Isotope-labeling measurement.** The isotope-labeling experiment was conducted by using $^{13}CO_2$ (isotope purity, 99% and chemical purity, 99.9%, Tokyo Gas Chemicals Co., Ltd.) as the carbon source. Typically, 10 mg of photocatalysts, 2 mM of $[Ru^{II}(bpy)_3]Cl_2 \cdot 6H_2O$, 10 mM of BIH, 30 mL of acetonitrile and 100 μL of water were loaded into the reaction cell. The protocol of $^{13}CO_2$ photoreduction was the same as that mentioned above. The gas products were analyzed by gas chromatography–mass spectrometry (JMS-K9, JEOL-GCQMS, Japan and 6890 N Network GC system, Agilent Technologies, USA) equipped with the column for detecting the products of $^{13}CO$ (HP-MOLESIEVE, 30 m × 0.32 mm × 25 μm). Helium was used as carrier gas. The column was maintained at 60 °C for 20 min, and the flow of the carrier was 0.5 ml L$^{-1}$. The temperatures of the injector, EI source, and the GCITF were set to be 200, 200, and 250 °C, respectively.

## Data availability

All data are available from the corresponding author on request. Source data are provided with this paper. Source data are also available in figshare with the identifier https://doi.org/10.6084/m9.figshare.12715484.

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

## Acknowledgements

This work was supported by NSFC (51932007, 21573170, U1705251, 51961135303, and 51902121), the National Key Research and Development Program of China (2018YFB1502001), National Postdoctoral Program for Innovative Talents (BX20190259), China Postdoctoral Science Foundation (2019M660189), and the Fundamental Research Funds for the Central Universities (WUT: 2019IVA111). J.X. is grateful to the financial support by the Australian Research Council. The project is also supported by the State Key Laboratory of Advanced Technology for Materials Synthesis and Processing (Wuhan University of Technology) (2018-KF-17).

## Author contributions

F.X., B.C., and J.Y. conceived and designed the experiments. F.X. and K.M. carried out the synthesis of the materials and the characterizations of the materials. F.X. and S.W. carried out the photocatalytic test. F.X., S.W., J.X., and J.Y. contributed to data analysis. J.Y. and J.X. supervised the project. F.X. wrote the paper. J.Y., S.W., and J.X. revised and reviewed the paper. All authors discussed the results and commented on the paper.

## Competing interests

The authors declare no competing interests.
