## [Peer Review File · Nature Communications]

Reviewers' comments:

Reviewer #1 (Remarks to the Author):

This work is interesting in the aspect of exploring the photocatalytic applications of perovskite material for CO₂ conversion by combining CsPbBr₃ QDs with a conventional semiconductor material TiO₂. The resultant promising activity prompted the mechanistic study that has revealed a S-scheme interface charge separation between the two components. Most of the discussion based on the experimental evidences is convincing. Some suggestions/questions are raised as follows for authors to consider in order further strengthen the fundamental aspect of this work.

1. What's the advantage of using the TiO₂ nanofibers in this work, since it's surface area and pore volume are quite low? It consists of both anatase and rutile phases, similar to commercial P25 TiO₂. Thus, the final photocatalyst contains heterojunctions (1) between QDs and TiO₂, and (2) between anatase and rutile TiO₂. The latter (2) is the majority since the percentage of QDs is low. How would the junctions between the anatase and rutile affect the charge separation between QDs and TiO₂?
2. For the proposed S-scheme interface charge transfer to occur, both TiO₂ and CsPbBr₃ QDs need to be activated by photon, which means UV light must be present during the reaction to be absorbed by TiO₂. The Xe lamp contains both UV and visible light. No bandpass filter was used in this study. Control study using visible light only should be conducted to provide further complementary evidence of the reaction mechanism.
3. It is well known that perovskite materials are unstable. In this work, recycle study was carried out and it was shown that the photocatalyst is rather stable. Characterization of the photocatalyst after reaction is suggested to understand the materials properties better.

Reviewer #2 (Remarks to the Author):

This manuscript describes the use TiO₂/CsPbBr₃ hybrid material for CO₂ photoreduction. Authors attribute the photocatalytic behaviour of this heterojunction to a S-scheme mechanism. To justify the photocatalytic performance and the photo-generated electron transfer dynamics a series of characterization studies have been also performed. However, several aspects related to the photocatalytic experiments and other linked issues are unclear.

I do not recommend the acceptance of this article in Nature communications. In order to justify this decision, I provide to the authors some comments and remark some that should be taking into account for its further publication.

1. One of the most interesting properties of halide perovskites is their high photoluminescence quantum yield. This behaviour is observed in the proposed materials in the figure 1e. However, high photoluminescence is counterproductive for photocatalytic experiments since it involves high recombination. Can you explain how this high PL affects photocatalytic behavior?
2. Authors attributed the changes in the textural properties (volcano shape) "because overloading of CsPbBr₃ QDs fill the nanopores of TiO₂". Taking into account the pore distribution (Supplementary Figure 3) TiO₂ have a pore diameter higher than 2 nm, therefore these are mesopores. Previous explanation is not of pore filling is also not supported because TC4 shows the same pore distribution than TiO₂, while TC2 exhibit an increase of pores with lower diameter. The isotherms of other TC materials should be included in the comparison (Supplementary Figure 3). In addition, units must be included in Supplementary Table 1.
3. Authors assign the changes in the band gaps to the interaction between CsPbBr₃ QDs and TiO₂. To confirm this interactions, they perform FTIR analysis showing changes in the wavenumber of organic groups (increasing with the QD amount) together with a decrease in OH signal. They justify the presence of chemical interactions. First, the decrease of OH is not easy to follow in the Figure 3 and may be also due to a coverage of TiO₂ by QDs. On the other hand, the changes in the wavenumber are negligible with the increase of QDs.
4. This behavior is also tried to justify by XPS. Where changes are detected in the bond energies of Ti, O, Br, Cs and Pb. In the case of Ti, Cs and Br, these variations are small, especially taking into account the width of the deconvolution peaks. In addition, they are assigned to the formation of Br-O bonds that in my opinion have no relationship with the comments in the FTIR section, where variations were observed in the organic part. To study these changes, you should include the C1s spectra.
5. As is described in the experimental section, photocatalytic experiments are performed in a flask of 50 ml. Previous irradiation the sample is bubbled with CO₂. However is not clear if the experiments have been performed in batch or continuous mode. This is important to evaluate the products quantification. The column used to detect the gasses must be provided. To detect the products, they are using a FID. However, how O₂ are measured? What happens with H₂? Authors explain that experiments are performed at ambient temperature, however the lamp of 350W is placed at 20 cm, therefore they will have an increase of the temperature. This should be also taking into account because they are also using acetonitrile in the liquid medium.
6. Regarding to product distribution, how is the O₂ production with the time? Are you sure that this is coming from the reaction? You must include the comparison with N₂ signal and the O₂ and the oxygen from the CO₂ cylinder that usually have between 40-80 ppm even for high purity.
7. The photocatalytic activity of bare QDs must be also included to evaluate the synergistic effect.
8. Taking into account your scheme (Figure 4d) light is only photogenerate electrons in TiO₂. How are generated the electron in QDs to perform the reduction reactions? Considering the QDs light absorption could you determine the filter effect over TiO₂?
9. How you perform the recyclability experiments? Did you remove the sample, clean it and reuse it? One of the main drawbacks of perovskite halides is their instability under water and UV light. XRD of used heterojunctions must be included in the paper discussion.
10. In the case of ¹³C label experiments is not clear how are performed. Did you use the same procedure previously described? Which is the detector? In figure 4 b only CO signals are presented.

What happen with CH₄ and CH₃OH? Together with the spectrum the chromatograms have to be included.

11. What happened with oxidation reaction? Is acetonitrile the electron donor? In this case you are producing similar amount of CO₂ (in the oxidation step) as the one you are transforming (in the reduction step).

12. On the other hand, you are using organic products in the synthesis of TiO₂ and QDs that are not totally removing (observed in FTIR). Previous studies have demonstrated that these carbonaceous products introduce uncertainties in the quantification of the reactions products. Elemental analysis and TG experiments must be included to determine the organic products presents in your materials.

13. Of course than CO₂ adsorption is an important step in their photoreduction. However, in the case of your DFT calculations only physisorption is observed with long distances form material surface make it difficult the charge transfers from the catalyst.

14. Authors attributed this behavior to a S-scheme, however this is not clear. PL spectra of heterojunctions show a decrease in the TiO₂ emission. However, this can also due to a filter effect. To corroborate this behavior time resolved PL experiments must be performed. In addition, even in the case of this happen the recombination of e⁻ (From TiO₂) with h⁺ (form QDs) only would only have an important effect if the reduction were carried out in the TiO₂.

Reviewer #3 (Remarks to the Author):

The authors present the synthesis and functional characterization of TiO₂ fibers decorated with CsPbBr₃ QDs for use as a CO₂ photoreduction catalyst. In this work, they characterize the structure of their composite materials by XPS, XRD, TEM, and FTIR and subsequently perform performance testing of their new material. They hypothesize the superior performance of their material is due to the formation of an S-type charge transfer mechanism. They support this claim through DFT calculations. This work in general is performed well and represents a significant improvement in the state-of-the-art based on table 1. There are a few points that need to be addressed before I can recommend publication in Nature Communications:

1. Structure of CsPbBr₃ QDs. The authors refer to the QDs as monoclinic but their XRD pattern shows additional peaks that are not described by this phase (~27, 39, 45 degrees 2theta). Can the authors comment on potential secondary phases? Additionally the structure shown in figure 1a is the cubic phase this is not consistent with the structure reported in 1d and should be corrected.
2. Similar to the above comment regarding the phase it appears that the calculations were performed using the cubic CsPbBr₃ phase rather than the monoclinic phase. These calculations should be redone with the correct phase. Since this is shown in support of the proposed S-type charge transfer the validity of this claim is concerning in its current form.
3. Pg. 17 lines 285-288, The sentence starting "The resulting...", is confusing and hard to understand. Please revise for clarity. The manuscript would also benefit from a careful read through for grammar errors.

-Laura Schelhas, SLAC National Accelerator Laboratory

Response to Reviewers' comments

Reviewer #1:

This work is interesting in the aspect of exploring the photocatalytic applications of perovskite material for CO₂ conversion by combining CsPbBr₃ QDs with a conventional semiconductor material TiO₂. The resultant promising activity prompted the mechanistic study that has revealed an S-scheme interface charge separation between the two components. Most of the discussion based on the experimental evidences is convincing. Some suggestions/questions are raised as follows for authors to consider in order further strengthen the fundamental aspect of this work.

1. What's the advantage of using the TiO₂ nanofibers in this work, since its surface area and pore volume are quite low? It consists of both anatase and rutile phases, similar to commercial P25 TiO₂. Thus, the final photocatalyst contains heterojunctions (1) between QDs and TiO₂, and (2) between anatase and rutile TiO₂. The latter (2) is the majority since the percentage of QDs is low. How would the junctions between the anatase and rutile affect the charge separation between QDs and TiO₂?

Response: In this work, we chose TiO₂ nanofibers as matrix to hybridize with CsPbBr₃ QDs for CO₂ photoreduction based on the following reasons: (i) TiO₂ nanofibres show no aggregation upon dispersion in solution and thereby retain their photocatalytical active sites exposed on the surface. (ii) Randomly stacked TiO₂ nanofibres readily form a loose network, facilitating the transports of gaseous reactants and products. (iii) The TiO₂ nanofibres are composed of small nanocrystals, possessing inter-particle voids and rough surface, which make TiO₂ nanofibres a favorable host to anchor CsPbBr₃ QDs.

According to our previous work (*Int. J. Hydrogen Energy*, 2014, 39, 15394-15402), anatase and rutile TiO₂ form S-scheme homojunction. As evidenced by DFT results (Figure 3d and e), the work function of anatase TiO₂ (101) was larger than that of rutile TiO₂ (110), indicating that electrons would flow from rutile to anatase and created an internal electric field (IEF) at anatase/rutile TiO₂ interfaces. Driven by the interfacial IEF, the photogenerated electrons in anatase TiO₂ CB spontaneously slid towards rutile TiO₂ VB and recombined with the holes in rutile TiO₂ VB. Such transportation of photoinduced charge carriers follows an S-like pathway (S-scheme homojunction) between anatase and rutile TiO₂ (Supplementary Figure 11 and Figure R1 below). When CsPbBr₃ QDs deposited on TiO₂ nanofibers, all possible schematic illustrations among anatase TiO₂, rutile TiO₂ and CsPbBr₃ QDs are shown in Supplementary Figure 12 and Figure R2 below. The corresponding discussions have been added in Page 17-18 in the revised manuscript.

Figure R1. Schematic illustration of anatase/rutile homojunction: internal electric field-induced charge transfer, separation and the formation of S-scheme heterojunction under UV-visible light irradiation.

Figure R2. Schematic illustrations of all possible $\text{TiO}_2/\text{CsPbBr}_3$ heterojunctions: (a)

CsPbBr₃ QDs contacted with anatase TiO₂, (b) CsPbBr₃ QDs contacted with rutile TiO₂, (c) CsPbBr₃ QDs contacted with anatase TiO₂ and anatase TiO₂ contacted with rutile TiO₂, (d) CsPbBr₃ QDs contacted with rutile TiO₂ and rutile TiO₂ contacted with anatase TiO₂.

2. For the proposed S-scheme interface charge transfer to occur, both TiO₂ and CsPbBr₃ QDs need to be activated by photon, which means UV light must be present during the reaction to be absorbed by TiO₂. The Xe lamp contains both UV and visible light. No bandpass filter was used in this study. Control study using visible light only should be conducted to provide further complementary evidence of the reaction mechanism.

Response: As you suggested, the photocatalytic activity of TiO₂/CsPbBr₃ was further evaluated under Xe lamp irradiation with a UV cutoff ($\lambda > 420$ nm). Under this condition, no photocatalytic products (CO, CH₄ or CH₃OH) were detected, agreeing with the Reviewer's suggestion and further verified the proposed reaction mechanism.

3. It is well known that perovskite materials are unstable. In this work, recycle study was carried out and it was shown that the photocatalyst is rather stable. Characterization of the photocatalyst after reaction is suggested to understand the materials properties better.

Response: Perovskite materials are normally unstable in humid air or in aqueous solution. However, unlike other reaction systems in which perovskite materials were in direct contact with water or oxygen, the photocatalytic CO₂ reduction in the revised manuscript was performed in gas atmosphere in a homemade container, where humid air was completely removed by high-purity N₂ flux prior to the experiment. The CO₂ gas was *in situ* produced through the reaction of NaHCO₃ powder and H₂SO₄ and diffused through a channel to the TiO₂/CsPbBr₃ photocatalysts (Figure R3). Under this condition, the perovskite QDs were not in contact with water/oxygen and thus remained stable after a few reaction cycles. Please refer to the Supporting Information for more experimental details.

Following your suggestion, we have characterized the recycled photocatalyst using XRD, TEM and XPS. As shown in the XRD pattern (Supplementary Figure 8a and Figure R4a), the used photocatalyst showed no detectable phase change. The TEM image shows that the QDs grew a bit larger after cycled photocatalytic reactions, while no significant aggregation was observed (Supplementary Figure 8b and Figure R4b). The chemical states of the used photocatalyst were also consistent with those of the fresh one, as examined by XPS (Supplementary Figure 9 and Figure R5 below). These results demonstrate the high chemical and structural stabilities of the TiO₂/CsPbBr₃ hybrid upon the gas-phase photocatalytic CO₂ reduction.

Figure R3. Homemade quartz reactor for photocatalytic CO₂ conversion.

Figure R4. (a) XRD patterns of TC2 before and after reaction. (b) TEM image of TC2 after reaction.

Figure R5. *Ex-situ* XPS spectra of (a) Cs 3d, (b) Pb 4f and (c) Br 3d of TC2 after reaction.

Reviewer #2:

This manuscript describes the use of $\text{TiO}_2/\text{CsPbBr}_3$ hybrid material for CO_2 photoreduction. Authors attribute the photocatalytic behaviour of this heterojunction to an S-scheme mechanism. To justify the photocatalytic performance and the photo-generated electron transfer dynamics, a series of characterization studies have been also performed. However, several aspects related to the photocatalytic experiments and other linked issues are unclear. I do not recommend the acceptance of this article in Nature communications. In order to justify this decision, I provide to the authors some comments and remark some that should be taken into account for its further publication.

1. One of the most interesting properties of halide perovskites is their high photoluminescence quantum yield. This behavior is observed in the proposed materials in the Figure 1e. However, high photoluminescence is counterproductive for photocatalytic experiments since it involves high recombination. Can you explain how this high PL affects photocatalytic behavior?

Response: It is worth pointing out that high photoluminescence indicates high *radiative* recombination between charge carriers, while *non-radiative* recombination does not give photoluminescence. Generally, strong photoluminescence indicates low defect states of the material, specifically perovskite QDs in the present work. Therefore, it is desirable to use the as-synthesized QDs to construct heterojunction with TiO_2 to improve the photocatalytic activity. In the $\text{TiO}_2/\text{CsPbBr}_3$ S-scheme heterostructure, there exists a built-in electric field at interfaces, which drives electrons from TiO_2 conduction band (CB) to combine with holes in CsPbBr_3 valence band (VB). The remaining electrons and holes thereby achieved a spatial separation, significantly promoting CO_2 photoreduction because of their stronger reduction ability in the S-scheme heterojunction.

2. Authors attributed the changes in the textural properties (volcano shape) “because overloading of CsPbBr_3 QDs fill the nanopores of TiO_2 ”. Taking into account the pore distribution (Supplementary Figure 3) TiO_2 have a pore diameter higher than 2 nm, therefore these are mesopores. Previous explanation is not of pore filling, is also not supported because TC4 shows the same pore distribution than TiO_2 , while TC2 exhibit an increase of pores with lower diameter. The isotherms of other TC materials should be included in the comparison (Supplementary Figure 3). In addition, units must be included in Supplementary Table 1.

Response: Following your suggestion, we have measured the N_2 sorption isotherms of other TC samples (Supplementary Figure 3 and Figure R6). All the TCx showed highly similar pore size distributions in a range of 10~20 nm, much larger than the

size of CsPbBr₃ QDs (6~9 nm). These results suggest that CsPbBr₃ QDs are very likely to fill into the pores of TiO₂.

The specific surface area (S_{BET}) derived from BET equation was rather reliable, since the correlation coefficient of BET fitting for each sample was higher than 0.999. The S_{BET} of each TC sample is presented in Supplementary Table S1 and below, which shows volcano-shape. At low QDs loadings (< 2 wt.%), TCx showed an increased S_{BET} and reached the maximum value at TC2. This is reasonable because the low filling enables QDs to deposit onto the inner wall of TiO₂ mesopores. Such island-like QDs on the inner walls may contribute additional specific surface area for the hybrid. When the QDs loading was further increased, QDs would aggregated in TiO₂ mesopores and the island-like distribution vanished, which thereby resulted in a decrease of S_{BET} .

According to the Reviewer's suggestion, we have updated the results and corresponding discussion in the revised manuscript.

Figure R6. Nitrogen adsorption-desorption isotherm and the corresponding pore size distribution (inset) of T, TC0.5, TC1, TC2, TC3 and TC4.

Table R1. Physical properties of the samples with different CsPbBr₃ QDs loadings.

Samples	S_{BET} (m ² /g)	V_{p} (m ³ /g)	d_{p} (nm)
T	19	0.09	17.6
TC0.5	21	0.09	18.4
TC1	28	0.14	20.2
TC2	42	0.15	17.1
TC3	22	0.08	14.5

TC4 18 0.07 13.0

S_{BET} : specific surface area, V_p : pore volume, d_p : average pore size

3. Authors assign the changes in the band gaps to the interaction between CsPbBr₃ QDs and TiO₂. To confirm these interactions, they perform FTIR analysis showing changes in the wavenumber of organic groups (increasing with the QD amount) together with a decrease in OH signal. They justify the presence of chemical interactions. First, the decrease of OH is not easy to follow in the Figure 3 and may be also due to a coverage of TiO₂ by QDs. On the other hand, the changes in the wavenumber are negligible with the increase of QDs.

Response: (i) We have zoomed in the FTIR spectra to highlight the variation of Ti-OH signal with the loading of QDs. As shown in Supplementary Figure 5 and Figure R7 below, the absorbance of (Ti)-OH at 3400 cm⁻¹ decreased with the increased loading of QDs, which should be ascribed to the chemical reaction between (Ti)-OH and QDs, *i.e.*, the formation of Ti-O-Br linkage.

(ii) The penetration depth of infrared radiation is in micrometer scale (*Analyst*, 2015, 140, 2093–2100), which is much larger than the size of QDs (6–9 nm). Therefore, the absorbance decrease of the (Ti)-OH band should not be assigned to the QDs coverage.

(iii) The absorption bands at 2950, 2920 and 2850 cm⁻¹ correspond to the vibrations of the C–H groups (*Adv. Mater.*, 2016, 28, 8718–8725), indicating the presence of organic groups on CsPbBr₃ QDs. The C–H stretching vibration band shifted 4 cm⁻¹ toward higher wavenumber upon hybridization with TiO₂, implying the interaction between QDs and TiO₂.

Figure R7. FTIR spectra of TiO₂, CsPbBr₃ and TCx.

4. This behavior is also tried to justify by XPS. Where changes are detected in the bond energies of Ti, O, Br, Cs and Pb. In the case of Ti, Cs and Br, these variations are

small, especially taking into account the width of the deconvolution peaks. In addition, they are assigned to the formation of Br-O bonds that in my opinion have no relationship with the comments in the FTIR section, where variations were observed in the organic part. To study these changes, you should include the C1s spectra.

Response: As we mentioned above, we believe the disappearance of the Ti-(OH) vibration band was due to the formation of Ti-O-Br bond, as indicated by FTIR. The C1s spectra of the samples were recorded and shown in Figure R8 below (also added as Figure S6 in Supporting Information). Only environmental carbon (binding energy 284.6 eV) was observed and all the three samples (TiO₂, TiO₂/CsPbBr₃ hybrid and CsPbBr₃) presented the same binding energy. These results are quite reasonable, since carbon was not involved in the TiO₂/CsPbBr₃ chemical interactions according to our suggestions.

Figure R8. *Ex-situ* XPS spectra of C 1s of pure TiO₂, CsPbBr₃ and TC2 nanohybrid.

5. As is described in the experimental section, photocatalytic experiments are performed in a flask of 50 ml. Previous irradiation the sample is bubbled with CO₂. However is not clear if the experiments have been performed in batch or continuous mode. This is important to evaluate the products quantification. The column used to detect the gasses must be provided. To detect the products, they are using a FID. However, how O₂ are measured? What happens with H₂? Authors explain that experiments are performed at ambient temperature, however the lamp of 350W is placed at 20 cm, therefore they will have an increase of the temperature. This should be also taking into account because they are also using acetonitrile in the liquid medium.

Response: The photocatalytic experiments were conducted in a batch mode. In the revised manuscript, we have adopted a gas-phase CO₂ photoreduction reaction for improved photocatalytic CO₂ reduction and to exclude the influence of acetonitrile on

the activity. The experimental details are as follows and added in the Supporting Information.

The photocatalytic reduction of CO₂ was carried out in a 200 mL home-made quartz reactor with two openings which were sealed using a silicone rubber septum. A 350 W Xe arc lamp (XD350, Changzhou Siyu, China) was used as the light source and positioned 10 cm above the photocatalytic reactor. In a typical photocatalytic experiment, 20 mg of the sample was put into the glass reactor and 10 mL of hexane was added. The catalyst was dispersed by ultrasonication for about 1 min to form suspension. After evaporation of hexane at 80 °C for 2 h, the sample was deposited on the bottom of the reactor in the form of thin films. Before irradiation, the reactor was purged with N₂ (99.9999%) for 30 min to remove air and ensure that the reaction system was under anaerobic conditions. CO₂ and H₂O vapor were *in situ* generated by the reaction of NaHCO₃ (0.084 g, introduced into the reactor before seal) and H₂SO₄ aqueous solution (0.3 mL, 2 M) which was introduced into the reactor using a syringe. The temperature in the reactor stabilized at 45 °C upon irradiation. 400 μL of mixed gas was taken from the reactor at given intervals (1 h) during the irradiation and used for gas component analysis by Shimadzu GC-2014C gas chromatograph (Japan) equipped with a flame ionized detector (FID), thermal conductivity detector (TCD) and a methanizer. Blank experiments were carried out in the absence of CO₂ or light irradiation to confirm that CO₂ and light were two key influencing elements for photocatalytic CO₂ reduction. Control experiments were also used to verify whether the carbon resource was derived from CO₂ or the catalyst itself.

H₂ was not detected in the reactions, indicating that the hydrogen evolution reaction was suppressed over the TiO₂/CsPbBr₃ heterostructure. The further cause is the absence of co-catalyst

6. Regarding to product distribution, how is the O₂ production with the time? Are you sure that this is coming from the reaction? You must include the comparison with N₂ signal and the O₂ and the oxygen from the CO₂ cylinder that usually have between 40-80 ppm even for high purity.

Response: As mentioned in the last response, a gas-phase CO₂ photoreduction reaction was employed in the revised manuscript. CO₂ was *in situ* generated through the reaction of NaHCO₃ and H₂SO₄, instead of the CO₂ cylinder during the previous process. The quartz reactor was purged by high-impurity N₂ (99.9999%) to completely remove the air. GC analysis showed that no O₂ was detected before the photocatalytic reaction started. Therefore, we are very confident that O₂ evolved from the reaction.

7. The photocatalytic activity of bare QDs must be also included to evaluate de

synergetic effect.

Response: Following the review's comment, we evaluated the photocatalytic CO₂ reduction activity of bare QDs. Similar to pure TiO₂, bare QDs (sample C) exhibited much lower photocatalytic reaction rate. The result and discussion are presented in Figure 4a and Figure R9 in the revised manuscript.

Figure R9. Photocatalytic activities of CO₂ reduction over TiO₂, TC and bare CsPbBr₃ QDs (sample C).

8. Taking into account your scheme (Figure 4d) light is only photogenerate electrons in TiO₂. How are generated the electron in QDs to perform the reduction reactions? Considering the QDs light absorption could you determine the filter effect over TiO₂?

Response: In fact, both TiO₂ and QDs were photo-excited and electrons were generated in QDs as well to perform the CO₂ reduction. We have modified the scheme (Figure 4g) in the revised manuscript for clarification.

Figure R10. Schematic illustration of TiO₂/CsPbBr₃ heterojunction: the formation of S-scheme heterojunction under UV-visible light irradiation for CO₂ photoreduction.

9. How you perform the recyclability experiments? Did you remove the sample, clean it and reuse it? One of the main drawbacks of perovskite halides is their instability

under water and UV light. XRD of used heterojunctions must be included in the paper discussion.

Response: We performed four circling runs to evaluate the recyclability of the sample. The experimental protocol was the same as the original photocatalytic CO₂ reduction (Supporting Information). In the first run, 400 μL of mixed gas was taken for component analysis with Shimadzu GC-2014C gas chromatograph at 1-hour time interval. The reactor was then dried at 60 °C in the vacuum oven for 1 h to remove possibly-absorbed CO₂, H₂O and products. The procedure was repeated for the second, third and fourth cycling runs. The sample was directly reused without further cleaning.

We have characterized the recycled photocatalyst using XRD, TEM and XPS. As demonstrated in our response to Question 3 of Reviewer 1, the XRD pattern (Supplementary Figure 8a and Figure R4a) of the recycled photocatalyst showed no detectable phase change. The TEM image confirms that the QDs did not show any obvious aggregation after cycled photocatalytic reactions and the morphology was well maintained (Supplementary Figure 8b and Figure R4b). The chemical states of the used photocatalyst were also consistent with those of the fresh one, as examined by XPS (Supplementary Figure 9 and Figure R5). These results demonstrate the high chemical and structural stabilities of the TiO₂/CsPbBr₃ hybrid upon the gas-phase photocatalytic CO₂ reduction.

10. In the case of ¹³C label experiments is not clear how are performed. Did you use the same procedure previously described? Which is the detector? In figure 4 b only CO signals are presented. What happen with CH₄ and CH₃OH? Together with the spectrum the chromatograms have to be included.

Response: ¹³CO₂ isotope tracer experiment was conducted to verify the carbon source of the products by using ¹³C isotope-labeled sodium bicarbonate (NaH¹³CO₃, Cambridge Isotope Laboratories Inc., USA) and H₂SO₄ aqueous solution for the photocatalysis examinations. After 1 h of photocatalytic reaction, 500 μL of the mixed gas was taken out from the reactor and examined by a gas chromatography-mass spectrometer (GC-MS) (6980N network GC system-5975 inert mass selective detector, Agilent Technologies, USA) to analyze the products.

We have double-checked the isotope experiments and the signals of ¹³CO, ¹³CH₄ and ¹³CH₃OH, together with the spectra of the chromatograms have been shown in Figure 4b, Supplementary Figure 7b and below. These results confirmed that the production of CO, CH₄ and CH₃OH was from the CO₂ reduction, rather than from carbon contaminations. Experimental details are shown in Supporting Information.

Figure R11. GC-MS spectra of CO, CH₄ and CH₃OH obtained from the photocatalytic reduction of ¹²CO₂ and ¹³CO₂ over TG0.5.

Figure R12. The spectra of the chromatograms of TC2 after irradiation for 1 hour with different carbon sources.

11. What happened with oxidation reaction? Is acetonitrile the electron donor? In this case you are producing similar amount of CO₂ (in the oxidation step) as the one you are transforming (in the reduction step).

Response: In the revised manuscript, we performed gas-phase photocatalytic reaction with CO₂/H₂O vapor (See Supporting Information), instead of previous liquid-phase reaction in acetonitrile. Under the updated condition, water acted as electron donor and reacts with photoinduced holes of TiO₂ valance band (VB) to produce O₂. The O₂ product was determined with use of GC-2014C equipped with TCD detector as shown in Figure 4a. In the reduction step, CO₂ as electron acceptor reacts with photoinduced electrons and converts into solar fuels (Figure 4a).

12. On the other hand, you are using organic products in the synthesis of TiO₂ and

QDs that are not totally removing (observed in FTIR). Previous studies have demonstrated that these carbonaceous products introduce uncertainties in the quantification of the reactions products. Elemental analysis and TG experiments must be included to determine the organic products presents in your materials.

Response: Following your suggestion, we have performed elemental analysis of T, TC2 and C with an elemental analyzer (Vario EL cube, German), which shows very small amounts of organic groups existed in the CsPbBr₃ QDs (Table. R2).

TG analysis was not applied for elemental analysis of the sample. On one hand, CsPbBr₃ QDs will lose weight during high-temperature heating especially in oxygen atmosphere, which possibly overlaps the weight loss of carbonaceous species. On the other hand, carbonaceous species only comes from CsPbBr₃ QDs. The low loading of QDs in TC2 enabled it impossible to accurately analyze the carbon content with TG.

Table R2. Elemental analysis of T, TC2 and C with an elemental analyzer.

	C (wt.%)	H (wt.%)	N (wt.%)
T	0.49	0.14	0.09
TC2	1.78	0.27	0.09
C	3.61	0.6	0.16

13. Of course than CO₂ adsorption is an important step in their photoreduction. However, in the case of your DFT calculations only physisorption is observed with long distances form material surface make it difficult the charge transfers from the catalyst.

Response: Charge transfer from the photocatalysts to CO₂ molecules can still occur even only physisorption exists, which has been extensively reported in literatures (*e.g. Angew. Chem. Int. Ed.*, 2019, 58 (4), 1134-1137; *Appl. Catal. B-Environ.*, 2019, 254, 270–282; *Small*, 2017, 13 (15), 1603938; *J. CO₂ Util.*, 2017, 21, 327-335.).

14. Authors attributed this behavior to a S-scheme, however this is not clear. PL spectra of heterojunctions show a decrease in the TiO₂ emission. However, this can also due to a filter effect. To corroborate this behavior time resolved PL experiments must be performed. In addition, even in the case of this happen the recombination of e⁻ (From TiO₂) with h⁺ (form QDs) only would only have an important effect if the reduction were carried out in the TiO₂.

Response: Following the reviewer's comment, the time-resolved photoluminescence (TRPL) experiments were performed to gain a deep insight into the charge-transfer dynamics (Supplementary Figure 13b and below). Compared with the pristine TiO₂,

TC2 showed shorter lifetime (τ_{av} , inset of Supplementary Figure 13b), demonstrating more efficient electron transfer at the interfaces of TiO₂/CsPbBr₃ nanohybrid.

Figure R13. TRPL spectra with corresponding fitting results of TiO₂ and TC2.

Reviewer #3:

The authors present the synthesis and functional characterization of TiO₂ fibers decorated with CsPbBr₃ QDs for use as a CO₂ photoreduction catalyst. In this work, they characterize the structure of their composite materials by XPS, XRD, TEM, and FTIR and subsequently perform performance testing of their new material. They hypothesize the superior performance of their material is due to the formation of an S-type charge transfer mechanism. They support this claim through DFT calculations. This work in general is performed well and represents a significant improvement in the state-of-the-art based on table 1. There are a few points that need to be addressed before I can recommend publication in Nature Communications:

1. Structure of CsPbBr₃ QDs. The authors refer to the QDs as monoclinic but their XRD pattern shows additional peaks that are not described by this phase (~27, 39, 45 degrees 2theta). Can the authors comment on potential secondary phases? Additionally the structure shown in figure 1a is the cubic phase this is not consistent with the structure reported in 1d and should be corrected.

Response: We deeply apologized for the typo: the as-prepared CsPbBr₃ QDs were identified as the cubic phase by the authors but was wrongly worded in the previous manuscript. In fact, all the discussion (including DFT calculation) was performed based on cubic phase CsPbBr₃.

2. Similar to the above comment regarding the phase it appears that the calculations were performed using the cubic CsPbBr₃ phase rather than the monoclinic phase.

These calculations should be redone with the correct phase. Since this is shown in support of the proposed S-type charge transfer the validity of this claim is concerning in its current form.

Response: As mentioned in the above response, the CsPbBr₃ was indeed cubic phase and the calculations were thus based on the correct crystal structure.

3. Pg. 17 lines 285-288, The sentence starting “The resulting...”, is confusing and hard to understand. Please revise for clarity. The manuscript would also benefit from a careful read through for grammar errors.

Response: The language has been modified in the revised manuscript.

Reviewers' comments:

Reviewer #1 (Remarks to the Author):

The authors have addressed my comments sufficiently. I have also looked at their responses to other reviewers' comments. In my option, this manuscript is publishable after they provide the raw GC data to show the evidences of O₂, CH₄, and methanol production, as requested by Reviewer 2.

Reviewer #2 (Remarks to the Author):

Reviewer #2:

Many thanks to the authors for the efforts to improve the quality of the paper to be published in a relevant journal as it Nature Communications.

I'm very happy to accept the paper if the clarify some point that are still unclear in this investigation.

1. One of the most interesting properties of halide perovskites is their high photoluminescence quantum yield. This behavior is observed in the proposed materials in the Figure 1e. However, high photoluminescence is counterproductive for photocatalytic experiments since it involves high recombination. Can you explain how this high PL affects can affect to photocatalytic behavior?

Response: It is worth pointing out that high photoluminescence indicates high radiative recombination between charge carriers, while non-radiative recombination does not give photoluminescence.

Generally, strong photoluminescence indicates low defects states of the material, specifically perovskite QDs in the present work.

Therefore, it is desirable to use the as-synthesized QDs to construct heterojunction with TiO₂ to improve the photocatalytic activity. In the TiO₂/CsPbBr₃ S-scheme heterostructure, there exists a built-in electric field at interfaces, which drives electrons from TiO₂ conduction band (CB) to combine with holes in CsPbBr₃ valance band (VB). The remaining electrons and holes thereby achieved a spatial separation, significantly promoting CO₂ photoreduction because of their stronger reduction ability in the S-scheme heterojunction.

Re-response: I agree with the authors that a high photoluminescence leads to a higher radiative recombination. However, if this luminescence is high and the lifetime of these photogenerated charges is lower than the required time for charge transfer, even these radiative recombination pathways are not beneficial for the photocatalytic processes. In the figure S13 the lifetime for TC2 is lower than for T. Which is the emission wavelength range used for this comparison?

2. Authors assign the changes in the band gaps to the interaction between CsPbBr₃ QDs and TiO₂. To confirm these interactions, they preform FTIR analysis showing changes in the wavenumber of organic groups (increasing with the QD amount) together with a decrease in OH signal. They justifying the presence of chemical interactions. First, the decrease of OH is not easy to follow in the Figure 3 and may be also due to a coverage of TiO₂ by QDs. On the other hand, the changes in the wavenumber are negligible with the increase of QDs.

Response: (i) We have zoomed in the FTIR spectra to highlight the variation of Ti-OH signal with the

loading of QDs. As shown in Supplementary Figure 5 and Figure R7 below, the absorbance of (Ti)-OH at 3400 cm^{-1} decreased with the increased loading of QDs, which should be ascribed to the chemical reaction between (Ti)-OH and QDs, i.e., the formation of Ti-O-Br linkage. (ii) The penetration depth of infrared radiation is in micrometer scale (Analyst, 2015, 140, 2093–2100), which is much larger than the size of QDs (6~9 nm).

Therefore, the absorbance decrease of the (Ti)-OH band should not be assigned to the QDs coverage.

(iii) The absorption bands at 2950 , 2920 and 2850 cm^{-1} correspond to the vibrations of the C–H groups (Adv. Mater., 2016, 28, 8718–8725), indicating the presence of organic groups on CsPbBr₃ QDs. The C–H stretching vibration band shifted 4 cm^{-1} toward higher wavenumber upon hybridization with TiO₂, implying the interaction between QDs and TiO₂.

Re-Response: I partially agree with the answer of the authors. Is true that IR analysis is a bulk technique if the experiments are performed in transmission or ATR, but not in the case of DRIFT. More information about that measurements must be provided. So, in the case that these experiments have been performed using Transmission or ATR (bilk) and taking into account that OH groups are in the surface, it would be difficult to determine the effect of QDs charge in the interaction with superficial OH.

On the other hand, which is the spectral resolution of your measurements? In most of the equipment including your IR, do not have sense decrease the spectral resolution below to 4 cm^{-1} . So, observed differences in your samples "...The C–H stretching vibration band shifted 4 cm^{-1} toward higher wavenumber upon hybridization with TiO₂..." is in the order of the technique error.

3. As is described in the experimental section, photocatalytic experiments are performed in a flask of 50 ml. Previous irradiation the sample is bubbled with CO₂. However is not clear if the experiments have been performed in batch or continuous mode. This is important to evaluate the products quantification. The column used to detect the gasses must be provided. To detect the products, they are using a FID. However, how O₂ are measured? What happens with H₂? Authors explain that experiments are performed at ambient temperature, however the lamp of 350W is placed at 20 cm, therefore they will have an increase of the temperature. This should be also taking into account because they are also using acetonitrile in the liquid medium.

Response: The photocatalytic experiments were conducted in a batch mode. In the revised manuscript, we have adopted a gas-phase CO₂ photoreduction reaction for improved photocatalytic CO₂ reduction and to exclude the influence of acetonitrile on the activity. The experimental details are as follows and added in the Supporting Information.

The photocatalytic reduction of CO₂ was carried out in a 200 mL home-made quartz reactor with two openings which were sealed using a silicone rubber septum. A 350 W Xe arc lamp (XD350, Changzhou Siyu, China) was used as the light source and positioned 10 cm above the photocatalytic reactor. In a typical photocatalytic experiment, 20 mg of the sample was put into the glass reactor and 10 mL of hexane was added. The catalyst was dispersed by ultrasonication for about 1 min to form suspension. After evaporation of hexane at 80 °C for 2 h, the sample was deposited on the bottom of the reactor in the form of thin films. Before irradiation, the reactor was purged with N₂ (99.9999%) for 30 min to remove air and ensure that the reaction system was under anaerobic conditions. CO₂ and H₂O vapor were in situ generated by the reaction of NaHCO₃ (0.084 g, introduced into the reactor before seal) and

H₂SO₄ aqueous solution (0.3 mL, 2 M) which was introduced into the reactor using a syringe. The temperature in the reactor stabilized at 45 °C upon irradiation. 400 µL of mixed gas was taken from the reactor at given intervals (1 h) during the irradiation and used for gas component analysis by Shimadzu GC-2014C gas chromatograph (Japan) equipped with a flame ionized detector (FID), thermal conductivity detector (TCD) and a methanizer. Blank experiments were carried out in the absence of CO₂ or light irradiation to confirm that CO₂ and light were two key influencing elements for photocatalytic CO₂ reduction. Control experiments were also used to verify whether the carbon resource was derived from CO₂ or the catalyst itself.

H₂ was not detected in the reactions, indicating that the hydrogen evolution reaction was suppressed over the TiO₂/CsPbBr₃ heterostructure. The further cause is the absence of co-catalyst

Response: As you mentioned to previous referee, the Perovskite are highly sensitive to water, but in the characterization of use samples you don't see any changes in their structure. So How can you explain the improved stability in your samples in presence of water.

4. In the case of ¹³C label experiments is not clear how are performed. Did you use the same procedure previously described? Which is the detector? In figure 4 b only CO signals are presented. What happen with CH₄ and CH₃OH? Together with the spectrum the chromatograms have to be included.

Response: ¹³CO₂ isotope tracer experiment was conducted to verify the carbon source of the products by using ¹³C isotope-labeled sodium bicarbonate (NaH¹³CO₃, Cambridge Isotope Laboratories Inc., USA) and H₂SO₄ aqueous solution for the photocatalysis examinations. After 1 h of photocatalytic reaction, 500 µL of the mixed gas was taken out from the reactor and examined by a gas chromatography-mass spectrometer (GC-MS) (6980N network GC system-5975 inert mass selective detector, Agilent Technologies, USA) to analyze the products. We have double-checked the isotope experiments and the signals of ¹³CO, ¹³CH₄ and ¹³CH₃OH, together with the spectra of the chromatograms have been shown in Figure 4b, Supplementary Figure 7b and below. These results confirmed that the production of CO, CH₄ and CH₃OH was from the CO₂ reduction, rather than from carbon contaminations. Experimental details are shown in Supporting Information. Figure R11. GC-MS spectra of CO, CH₄ and CH₃OH obtained from the photocatalytic reduction of ¹²CO₂ and ¹³CO₂ over TG0.5.

Re-Response. In order to confirm these results the separation column an analysis temperature must be also provided because the separation of CO from other gases is a tricky procedure.

5. What happened with oxidation reaction? Is acetonitrile the electron donor? In this case you are producing similar amount of CO₂ (in the oxidation step) as the one you are transforming (in the reduction step).

Response: In the revised manuscript, we performed gas-phase photocatalytic reaction with CO₂/H₂O vapor (See Supporting Information), instead of previous liquid-phase reaction in acetonitrile. Under the updated condition, water acted as electron donor and reacts with photoinduced holes of TiO₂ valance band (VB) to produce O₂. The O₂ product was determined with use of GC-2014C equipped with TCD detector as shown in Figure 4a. In the reduction step, CO₂ as electron acceptor reacts with photoinduced electrons and converts into solar fuels (Figure 4a).

Re-response: As I commented previously, and taking onto account that in this new experiments water is

used as electron donor. How you can explain the stability of your perovskite in presence of water?

6. On the other hand, you are using organic products in the synthesis of TiO₂ and QDs that are not totally removing (observed in FTIR). Previous studies have demonstrated that these carbonaceous products introduce uncertainties in the quantification of the reactions products. Elemental analysis and TG experiments must be included to determine the organic products presents in your materials.

Response: Following your suggestion, we have performed elemental analysis of T, TC2 and C with an elemental analyzer (Vario EL cube, German), which shows very small amounts of organic groups existed in the CsPbBr₃ QDs (Table. R2). TG analysis was not applied for elemental analysis of the sample. On one hand, CsPbBr₃ QDs will lose weight during high-temperature heating especially in oxygen atmosphere, which possibly overlaps the weight loss of carbonaceous species. On the other hand, carbonaceous species only comes from CsPbBr₃ QDs. The low loading of QDs in TC2 enabled it impossible to accurately analyze the carbon content with TG.

Re-Response: Elemental analysis is very useful in this case. To estimate the effect of these residues in the photocatalytic activity. Please calculate the which would be the production if all of these C-products will be transformed in reaction products.

7. Authors attributed this behavior to a S-scheme, however this is not clear. PL spectra of heterojunctions show a decrease in the TiO₂ emission. However, this can also due to a filter effect. To corroborate this behavior time resolved PL experiments must be performed. In addition, even in the case of this happen the recombination of e⁻ (From TiO₂) with h⁺ (form QDs) only would only have an important effect if the reduction were carried out in the TiO₂.

Response: Following the reviewer's comment, the time-resolved photoluminescence (TRPL) experiments were performed to gain a deep insight into the charge-transfer dynamics (Supplementary Figure 13b and below). Compared with the pristine TiO₂, TC2 showed shorter lifetime (τ_{av} , inset of Supplementary Figure 13b), demonstrating more efficient electron transfer at the interfaces of TiO₂/CsPbBr₃ nanohybrid.

Re-Response: As I commented you in a previous answer to explain this behavior is necessary to know which is the range (of emission wavelengths) that you are using in your measurements.

Reviewer #3 (Remarks to the Author):

The authors have addressed my previous concerns and I can now recommend this work for publication in Nature Communications.

Response to Reviewers' comments

Reviewer #1:

The authors have addressed my comments sufficiently. I have also looked at their responses to other reviewers' comments. In my option, this manuscript is publishable after they provide the raw GC data to show the evidences of O₂, CH₄, and methanol production, as requested by Reviewer 2.

Reviewer #2:

Many thanks to the authors for the efforts to improve the quality of the paper to be published in a relevant journal as it Nature Communications.

I'm very happy to accept the paper if the clarify some point that are still unclear in this investigation.

1. One of the most interesting properties of halide perovskites is their high photoluminescence quantum yield. This behavior is observed in the proposed materials in the Figure 1e. However, high photoluminescence is counterproductive for photocatalytic experiments since it involves high recombination. Can you explain how this high PL affects can affect to photocatalytic behavior?

Response: It is worth pointing out that high photoluminescence indicates high radiative recombination between charge carriers, while non-radiative recombination does not give photoluminescence. Generally, strong photoluminescence indicates low defects states of the material, specifically perovskite QDs in the present work.

Therefore, it is desirable to use the as-synthesized QDs to construct heterojunction with TiO₂ to improve the photocatalytic activity. In the TiO₂/CsPbBr₃ S-scheme heterostructure, there exists a built-in electric field at interfaces, which drives electrons from TiO₂ conduction band (CB) to combine with holes in CsPbBr₃ valance band (VB). The remaining electrons and holes thereby achieved a spatial separation, significantly promoting CO₂ photoreduction because of their stronger reduction ability in the S-scheme heterojunction.

Re-response: I agree with the authors that a high photoluminescence leads to a higher radiative recombination. However, if this luminescence is high and the lifetime of these photogenerated charges is lower than the required time for charge transfer, even these radiative recombination pathways are not beneficial for the photocatalytic

processes. In the figure S13 the lifetime for TC2 is lower than for T. Which is the emission wavelength range used for this comparison?

Response: The lifetime was monitored at emission wavelength of 480 nm for comparison. Based on the photocatalytic performances, we believe that at least to some extent, the lifetime of the photo-generated electrons is long enough for charge transfer. Nevertheless, detailed study on the lifetimes of charge carriers for recombination, trapping, interfacial transfer, *etc.*, is out of the scope of this manuscript.

Besides, as we have mentioned in the manuscript, the S-scheme $\text{TiO}_2/\text{CsPbBr}_3$ heterostructure was formed in the hybridized semiconductors. The S-scheme enabled an internal electric field at interfaces, which drove electrons in TiO_2 CB and holes in CsPbBr_3 VB to combine, achieving more efficient charge separation and high redox capability of the remaining charge carriers.

2. Authors assign the changes in the band gaps to the interaction between CsPbBr_3 QDs and TiO_2 . To confirm these interactions, they perform FTIR analysis showing changes in the wavenumber of organic groups (increasing with the QD amount) together with a decrease in OH signal. They justify the presence of chemical interactions. First, the decrease of OH is not easy to follow in the Figure 3 and may be also due to a coverage of TiO_2 by QDs. On the other hand, the changes in the wavenumber are negligible with the increase of QDs.

Response: (i) We have zoomed in the FTIR spectra to highlight the variation of Ti-OH signal with the loading of QDs. As shown in Supplementary Figure 5 and Figure R7 below, the absorbance of (Ti)-OH at 3400 cm^{-1} decreased with the increased loading of QDs, which should be ascribed to the chemical reaction between (Ti)-OH and QDs, i.e., the formation of Ti-O-Br linkage. (ii) The penetration depth of infrared radiation is in micrometer scale (Analyst, 2015, 140, 2093–2100), which is much larger than the size of QDs (6–9 nm).

Therefore, the absorbance decrease of the (Ti)-OH band should not be assigned to the QDs coverage.

(iii) The absorption bands at 2950 , 2920 and 2850 cm^{-1} correspond to the vibrations of the C–H groups (Adv. Mater., 2016, 28, 8718–8725), indicating the presence of organic groups on CsPbBr_3 QDs. The C–H stretching vibration band shifted 4 cm^{-1} toward higher wavenumber upon hybridization with TiO_2 , implying the interaction between QDs and TiO_2 .

Re-Response: I partially agree with the answer of the authors. It is true that IR analysis is a bulk technique if the experiments are performed in transmission or ATR, but not in the case of DRIFT. More information about that measurements must be provided. So, in the case that these experiments have been performed using Transmission or ATR (bulk) and taking into account that OH groups are in the surface,

it would be difficult to determine the effect of QDs charge in the interaction with superficial OH.

On the other hand, which is the spectral resolution of your measurements? In most of the equipment including your IR, do not have sense decrease the spectral resolution below to 4 cm⁻¹. So, observed differences in your samples "...The C–H stretching vibration band shifted 4 cm⁻¹ toward higher wavenumber upon hybridization with TiO₂..." is in the order of the technique error.

Response: (i) The FTIR spectra were measured with an attenuated total reflectance (ATR) mode on Nicolet iS 50 (Thermo fisher, USA). Each spectrum was collected by merging 32 scans with a 4 cm⁻¹ resolution. Calibration was conducted using a standard polystyrene sample prior to measurement. The IR measurement was repeated for 3 times on each sample, to exclude any possible technique error. Therefore, we believe that the IR shift of 4 cm⁻¹ is reliable.

(ii) The XPS analyses show that the amount of (Ti)-OH for TiO₂ decreased after hybridization with QDs, consistent with the FTIR result. Both XPS and FTIR results prove the chemical interaction between (Ti)-OH and CsPdBr₃ QDs, *i.e.*, the formation of Ti-O-Br linkage.

3. As is described in the experimental section, photocatalytic experiments are performed in a flask of 50 ml. Previous irradiation the sample is bubbled with CO₂. However is not clear if the experiments have been performed in batch or continuous mode. This is important to evaluate the products quantification. The column used to detect the gasses must be provided. To detect the products, they are using a FID. However, how O₂ are measured? What happens with H₂? Authors explain that experiments are performed at ambient temperature, however the lamp of 350W is placed at 20 cm, therefore they will have an increase of the temperature. This should be also taking into account because they are also using acetonitrile in the liquid medium.

Response: The photocatalytic experiments were conducted in a batch mode. In the revised manuscript, we have adopted a gas-phase CO₂ photoreduction reaction for improved photocatalytic CO₂ reduction and to exclude the influence of acetonitrile on the activity. The experimental details are as follows and added in the Supporting Information.

The photocatalytic reduction of CO₂ was carried out in a 200 mL home-made quartz reactor with two openings which were sealed using a silicone rubber septum. A 350 W Xe arc lamp (XD350, Changzhou Siyu, China) was used as the light source and positioned 10 cm above the photocatalytic reactor. In a typical photocatalytic experiment, 20 mg of the sample was put into the glass reactor and 10 mL of hexane was added. The catalyst was dispersed by ultrasonication for about 1 min to form

suspension. After evaporation of hexane at 80 °C for 2 h, the sample was deposited on the bottom of the reactor in the form of thin films. Before irradiation, the reactor was purged with N₂ (99.9999%) for 30 min to remove air and ensure that the reaction system was under anaerobic conditions. CO₂ and H₂O vapor were *in situ* generated by the reaction of NaHCO₃ (0.084 g, introduced into the reactor before seal) and H₂SO₄ aqueous solution (0.3 mL, 2 M) which was introduced into the reactor using a syringe.

The temperature in the reactor stabilized at 45 °C upon irradiation. 400 µL of mixed gas was taken from the reactor at given intervals (1 h) during the irradiation and used for gas component analysis by Shimadzu GC-2014C gas chromatograph (Japan) equipped with a flame ionized detector (FID), thermal conductivity detector (TCD) and a methanizer. Blank experiments were carried out in the absence of CO₂ or light irradiation to confirm that CO₂ and light were two key influencing elements for photocatalytic CO₂ reduction. Control experiments were also used to verify whether the carbon resource was derived from CO₂ or the catalyst itself.

H₂ was not detected in the reactions, indicating that the hydrogen evolution reaction was suppressed over the TiO₂/CsPbBr₃ heterostructure. The further cause is the absence of co-catalyst

Re-Response: As you mentioned to previous referee, the Perovskite are highly sensitive to water, but in the characterization of use samples you don't see any changes in their structure. So How can you explain the improved stability in your samples in presence of water.

Response: (i) The photocatalytic CO₂ reduction was performed in a gas atmosphere in a homemade container, where oxygen was completely removed by high-purity N₂ flux prior to the experiment. The CO₂ gas was *in situ* produced through the reaction of NaHCO₃ powder and 2 M of H₂SO₄, and the formed gas reagents diffused through a channel to the photocatalysts (Figure R1). Under this condition, the perovskite QDs were not in contact with liquid water and thus remained reasonably stable even after a few cycles.

(ii) The atmosphere in the system contains ~3% H₂O and ~97% N₂/CO₂, considering that the equilibrium pressure of pure water at 25 °C is 3.169 kPa. The mixed N₂/CO₂ gas as the major component was nondestructive to CsPbBr₃.

(iii) Previous work suggested that CsPbBr₃ can be stabilized by hydrophobic organic groups [*Nat. Energy*, 1 (2016), 15016.]. The CsPbBr₃ QDs here were terminated by the residual oleic acid species, as evidenced by FTIR results (Figure S5) and elemental analyses (Table R1).

(iv) Normally, "inorganic" perovskites such as CsPbBr₃ are usually more stable than "organic" perovskites (e.g. CH₃NH₃PbX₃). The inorganic perovskites demonstrate reasonable stability against moisture and oxygen [*Nano Lett.*, 17 (2017)

6759; *Adv. Mater.*, 27 (2015), 7101; *J. Phys. Chem. Lett.*, 10 (2019), 1217.].

Figure R1. Homemade quartz reactor for photocatalytic CO₂ conversion.

Table R1. Elemental analysis of T, TC2 and C with an elemental analyzer.

	C (wt.%)	H (wt.%)	N (wt.%)
T	0.49	0.14	0.09
TC2	1.78	0.27	0.09
C	3.61	0.6	0.16

4. In the case of ¹³C label experiments is not clear how are performed. Did you use the same procedure previously described? Which is the detector? In figure 4 b only CO signals are presented. What happen with CH₄ and CH₃OH? Together with the spectrum the chromatograms have to be included.

Response: ¹³CO₂ isotope tracer experiment was conducted to verify the carbon source of the products by using ¹³C isotope-labeled sodium bicarbonate (NaH¹³CO₃, Cambridge Isotope Laboratories Inc., USA) and H₂SO₄ aqueous solution for the photocatalysis examinations. After 1 h of photocatalytic reaction, 500 μL of the mixed gas was taken out from the reactor and examined by a gas chromatography-mass spectrometer (GC-MS) (6980N network GC system-5975 inert mass selective detector, Agilent Technologies, USA) to analyze the products. We have double-checked the isotope experiments and the signals of ¹³CO, ¹³CH₄ and ¹³CH₃OH, together with the spectra of the chromatograms have been shown in Figure 4b, Supplementary Figure 7b and below. These results confirmed that the production of CO, CH₄ and CH₃OH was from the CO₂ reduction, rather than from carbon contaminations. Experimental details are shown in Supporting Information. Figure R11. GC-MS spectra of CO, CH₄ and CH₃OH obtained from the photocatalytic reduction of ¹²CO₂ and ¹³CO₂ over TG0.5.

Re-Response: In order to confirm these results the separation column an analysis

temperature must be also provided because the separation of CO from other gases is a tricky procedure.

Response: The GC–MS analyses were carried out on a gas chromatography-mass spectrometer (6890N network GC system-5975 inert mass selective detector, Agilent Technologies, USA) equipped with an Agilent HP-5 capillary column (30.0 m × 0.32 mm i.d., film thickness 0.25 μm). Helium was used as the carrier gas. The column was maintained at 50 °C for 1 min and then heated to 100 °C at a programming rate of 8 °C min⁻¹. The temperature of injector was set to be 250 °C.

5. What happened with oxidation reaction? Is acetonitrile the electron donor? In this case you are producing similar amount of CO₂ (in the oxidation step) as the one you are transforming (in the reduction step).

Response: In the revised manuscript, we performed gas-phase photocatalytic reaction with CO₂/H₂O vapor (See Supporting Information), instead of previous liquid-phase reaction in acetonitrile. Under the updated condition, water acted as electron donor and reacts with photoinduced holes of TiO₂ valance band (VB) to produce O₂. The O₂ product was determined with use of GC-2014C equipped with TCD detector as shown in Figure 4a. In the reduction step, CO₂ as electron acceptor reacts with photoinduced electrons and converts into solar fuels (Figure 4a).

Re-Response: As I commented previously, and taking onto account that in this new experiments water is used as electron donor. How you can explain the stability of your perovskite in presence of water?

Response: Please refer to our response to your 3rd comment.

6. On the other hand, you are using organic products in the synthesis of TiO₂ and QDs that are not totally removing (observed in FTIR). Previous studies have demonstrated that these carbonaceous products introduce uncertainties in the quantification of the reactions products. Elemental analysis and TG experiments must be included to determine the organic products presents in your materials.

Response: Following your suggestion, we have performed elemental analysis of T, TC2 and C with an elemental analyzer (Vario EL cube, German), which shows very small amounts of organic groups existed in the CsPbBr₃ QDs (Table. R2). TG analysis was not applied for elemental analysis of the sample. On one hand, CsPbBr₃ QDs will lose weight during high-temperature heating especially in oxygen atmosphere, which possibly overlaps the weight loss of carbonaceous species. On the other hand, carbonaceous species only comes from CsPbBr₃ QDs. The low loading of QDs in TC2 enabled it impossible to accurately analyze the carbon content with TG.

Re-Response: Elemental analysis is very useful in this case. To estimate the effect of these residues in the photocatalytic activity. Please calculate that which would be the

production if all of these C-products will be transformed in reaction products.

Response: We have calculated the amount of C₁ products (CH₄, CO, CH₃OH, etc) from the organic residues, provided that all the carbon-containing residues were transformed, was only 0.03 mmol.

On the other hand, we would like to highlight again that, control experiment was carried out in the absence of CO₂ gas source to study whether the photocatalytic products were derived from CO₂ or the organic residues. No product can be detected under this condition, confirming that all the products originated from CO₂ reduction. Furthermore, the ¹³CO₂ isotope tracer experiments also confirmed the C₁ products were exclusively from the CO₂ gas.

7. Authors attributed this behavior to a S-scheme, however this is not clear. PL spectra of heterojunctions show a decrease in the TiO₂ emission. However, this can also due to a filter effect. To corroborate this behavior time resolved PL experiments must be performed. In addition, even in the case of this happen the recombination of e⁻ (From TiO₂) with h⁺ (from QDs) only would only have an important effect if the reduction were carried out in the TiO₂.

Response: Following the reviewer's comment, the time-resolved photoluminescence (TRPL) experiments were performed to gain a deep insight into the charge-transfer dynamics (Supplementary Figure 13b and below). Compared with the pristine TiO₂, TC2 showed shorter lifetime (τ_{av} , inset of Supplementary Figure 13b), demonstrating more efficient electron transfer at the interfaces of TiO₂/CsPbBr₃ nanohybrid.

Re-Response: As I commented you in a previous answer to explain this behavior is necessary to know which is the range (of emission wavelengths) that you are using in your measurements.

Response: Please refer to our response to your 1st comment.

Reviewer #3:

The authors have addressed my previous concerns and I can now recommend this work for publication in Nature Communications.

Reviewers' comments:

Reviewer #2 (Remarks to the Author):

In this new response the authors doesn't answer the most of the questions in a clear way. Some additional experiments have been performed such as Elemental analysis, but even in this case the answer is not correct. In addition, additional information shows several uncertainty in different experiments that not clarify the proposed questions with several unclear points (13C-labeled experiments, FTIR measurements, Carbon residues contribution, Photoluminescence lifetime experiments.)

So, I can't accept the paper in this way and I carefully recommend to the authors that read carefully the questions and answered it to no include more noise in this complex reaction.

1. Response: The lifetime was monitored at emission wavelength of 480 nm for comparison. Based on the photocatalytic performances, we believe that at least to some extent, the lifetime of the photo-generated electrons is long enough for charge transfer. Nevertheless, detailed study on the lifetimes of charge carriers for recombination, trapping, interfacial transfer, etc, is out of the scope of this manuscript. Besides, as we have mentioned in the manuscript, the S-scheme TiO₂/CsPbBr₃ heterostructure was formed in the hybridized semiconductors. The S-scheme enabled an internal electric field at interfaces, which drove electrons in TiO₂ CB and holes in CsPbBr₃ VB to combine, achieving more efficient charge separation and high redox capability of the remaining charge carriers.

Re-Response: I'm not in agreement with the author answer. First, the study of charge transfer carriers is not out of the scope when you want to demonstrate that has a S-type or other charge transfer scheme. Second, if you have performed the lifetime PL experiments at 480 you are only comparing the lifetimes of a TiO₂ luminescence signal. When QDs are deposited over TiO₂ a decrease in the luminescence lifetime is observed indicating a higher recombination rate. Therefore, this doesn't mean that you are... "achieving more efficient charge separation and high redox capability of the remaining charge carriers". In order to demonstrate that "...an internal electric field at interfaces, which drove electrons in TiO₂ CB and holes in CsPbBr₃ VB..." you must include the lifetime at 520 nm (QD emission). If you observe an increase in the lifetime these could mean that TiO₂ is transferring electrons in TiO₂ CB and holes in CsPbBr₃ VB.

So, please revise this part of the manuscript, repeat the experiments using a wavelength source distinctive for QDs and change your explanation because these data are not consistent with the conclusions.

2. Response: (i) The FTIR spectra were measured with an attenuated total reflectance (ATR) mode on Nicolet IS 50 (Thermo fisher, USA). Each spectrum was collected by merging 32 scans with a 4 cm⁻¹ resolution. Calibration was conducted using a standard polystyrene sample prior to measurement. The IR measurement was repeated for 3 times on each sample, to exclude any possible technique error. Therefore, we believe that the IR shift of 4 cm⁻¹ is reliable. (ii) The XPS analyses show that the amount of (Ti)-OH for TiO₂ decreased after hybridization with QDs, consistent with the FTIR result. Both XPS and

FTIR results prove the chemical interaction between (Ti)-OH and CsPbBr₃ QDs, i.e., the formation of Ti-O-Br linkage.

Re-Response. (i) Regarding to FTIR (ATR). First, -OH region of TiO₂ is very strange, hydrosil group in TiO₂ exhibit a sharp peak at higher wavenumber than water molecules (which should have a high intensity and broad), but in your case this peak is almost non-existent. The disappearance of this “peak” can be also attributed to the deposition of QD particles over the TiO₂ surface (but don't demonstrate the interaction). In addition, not important changes are observed on the wavenumber of -OH signal to conclude an interaction with QDs. The changes in the intensity of the broad band from 3000 to 3500 cm⁻¹ are due to variations in the humidity of the sample. (ii) On the other hand, the changes in the XPS O1s signal are also unclear. Revising carefully the O1s signal of XPS experiments some assignments are not well defined. In the case of O1s peak of TiO₂, authors ascribed a broad ca 528-534 eV to -OH group. This is not totally true. Even when this contribution is difficult to convolute, this is due to oxygenates (that also include to adsorbed water of other oxygenate organic compounds). So the decrease of this signal can be also attributed to an increase in of QD over the TiO₂ surface or even to a decrease in the adsorbed water (which is in agreement with the H₂O broad band decrease in FTIR). But to conclude that it is primarily due to the interaction of QDs with the OH groups of TiO₂ cannot be firmly affirmed. Revise both part of the manuscript, the XPS a FTIR assignment and of course the conclusion, because with this data is not possible to conclude a chemical interaction between QDs and TiO₂.

3. Response: (i) The photocatalytic CO₂ reduction was performed in a gas atmosphere in a homemade container, where oxygen was completely removed by high-purity N₂ flux prior to the experiment. The CO₂ gas was in situ produced through the reaction of NaHCO₃ powder and 2 M of H₂SO₄, and the formed gas reagents diffused through a channel to the photocatalysts (Figure R1). Under this condition, the perovskite QDs were not in contact with liquid water and thus remained reasonably stable even after a few cycles.

(iv) Normally, “inorganic” perovskites such as CsPbBr₃ are usually more stable than “organic” perovskites (e.g. CH₃NH₃PbX₃). The inorganic perovskites demonstrate reasonable stability against moisture and oxygen [Nano Lett., 17 (2017) 6759; Adv. Mater., 27 (2015), 7101; J. Phys. Chem. Lett., 10 (2019), 1217.].

Figure R1. Homemade quartz reactor for photocatalytic CO₂ conversion.

Response: (i) The photocatalytic CO₂ reduction was performed in a gas atmosphere in a homemade container, where oxygen was completely removed by high-purity N₂ flux prior to the experiment. The CO₂ gas was in situ produced through the reaction of NaHCO₃ powder and 2 M of H₂SO₄, and the formed gas reagents diffused through a channel to the photocatalysts (Figure R1). Under this condition, the perovskite QDs were not in contact with liquid water and thus remained reasonably stable even after a few cycles.

(ii) The atmosphere in the system contains ~3% H₂O and ~97% N₂/CO₂, considering that the equilibrium pressure of pure water at 25 °C is 3.169 kPa. The mixed N₂/CO₂ gas as the major component was nondestructive to CsPbBr₃.

(iii) Previous work suggested that CsPbBr₃ can be stabilized by hydrophobic organic groups [Nat. Energy,

1 (2016), 15016.]. The CsPbBr₃ QDs here were terminated by the residual oleic acid species, as evidenced by FTIR results (Figure S5) and elemental analyses (Table R1).

Table R1. Elemental analysis of T, TC2 and C with an elemental analyzer.

C (wt.%) H(wt.%) N (wt.%)

T 0.49 0.14 0.09

TC2 1.78 0.27 0.09

C 3.61 0.6 0.16

Re-response:

(i) The only way to eliminate the all O₂ in a glass reactor is using vacuum pumps, performing cold/heat cycles (with N₂ liquid) and using vacuum joints. In addition, even in the case that you are using high purity gases (99.9999%) these N₂ coming from de air so you have O₂ in you cylinders (some ppms) which are highly important in the recombination reactions. Moreover, your H₂SO₄ solution has O₂ solved. So you cannot affirm that O₂ completely removed from the reactor, may be you cannot measured. In addition, it is still not explained with is the column that you are using to determine the O₂. To avoid uncertain, please include a comparison with the N₂/O₂ ratio.

On the other hand, if your joints are not enough precise you could have a leak and this could be a possible explanation to not detect H₂. Other possibility to not detect H₂ is if you are using He as carrier gas in the GC. Even when major component is N₂ and CO₂, H₂O and O₂ or even H₂SO₄ vapor can destroy your sample.

(ii) I partially agree with your answer. It is possible that these compound increase the hydrophobic character of CsPbBr₃ QDs but could have other negative effects in the photocatalytic performance. In addition, the C percentages measured by Elemental Analysis are too high taking into account your production yields. To avoid uncertain include in a balance of the C-based products (CH₄, CH₃OH and CO) if only come from these carbon residues, and compare with your results. The calculation offer in answer 6 is not finessed (revise my Re-response to this question).

(iii) In addition to assure the answer (ii) and (iii) FTIR after reaction must be included to corroborate the presence of organic residues.

4. Response: The GC–MS analyses were carried out on a gas chromatography-mass spectrometer (6890N network GC system-5975 inert mass selective detector, Agilent Technologies, USA) equipped with an Agilent HP-5 capillary column (30.0 m × 0.32 mm i.d., film thickness 0.25 μm). Helium was used as the carrier gas. The column was maintained at 50 °C for 1 min and then heated to 100 °C at a programming rate of 8 °C min⁻¹. The temperature of injector was set to be 250 °C.

Re-Response. The Column (HP-5) is not adequate to separate permanent gases (such as H₂, CO, O₂, CH₄...). In addition the operation program including the starting temperature (50°C) is too high for gas separation. To avoid incertitude, please include the chromatogram of these experiments Furthermore, in the Figure S7b only CO₂ signals are shown. What's happened with products??

5. Response: Please refer to our response to your 3rd comment

Re-response: This answer is not completely response in 3rd comment. Please include FTIR spectra after

reaction.

6. Response: We have calculated the amount of C1 products (CH₄, CO, CH₃OH, etc) from the organic residues, provided that all the carbon-containing residues were transformed, was only 0.03 mmol. On the other hand, we would like to highlight again that, control experiment was carried out in the absence of CO₂ gas source to study whether the photocatalytic products were derived from CO₂ or the organic residues. No product can be detected under this condition, confirming that all the products originated from CO₂ reduction

Furthermore, the ¹³CO₂ isotope tracer experiments also confirmed the C1 products were exclusively from the CO₂ gas.

Re-Response. Many articles have shown that even only in presence of TiO₂ and in absence of CO₂ different products are observed (CO, CH₄...) mainly due to the presence of Carbonates or organic residues in the catalyst surface. Which is the difference in your case?

Regarding to the ¹³CO₂ experiments as I commented before some uncertain are still not solved.

In the case of your calculations of carbon-containing residues. This must be compared with your photocatalytic results. Taking into account the measured %C in your catalyst (1.78 %wt) you have 0.03 mmols or C. However your results are in micromole/g, so this means that you if you use 0.002 g in the experiments the C-based products ca be 1483.34 micromoles per gram, which is higher than your accumulative production even if you consider the summation of all recycled experiments.

7. Response: Please refer to our response to your 1st comment.

Re-response. As I commented in 1st comment this question is not fully addressed. The decrease on Fluorescence lifetime means that the recombination rate is increased. You are focusing only in the TiO₂ signal (480 nm), but you don't know where go the electrons. If yo see an increase in the Florescence of QDs or in his florescence lifetime you could assign these changes to a possible charge transfer from TiO₂ to QDs.

Response to Reviewers' comments

Reviewer #2:

In this new response the authors doesn't answer the most of the questions in a clear way. Some additional experiments have been performed such as Elemental analysis, but even in this case the answer is not correct. In addition, additional information shows several incertitude in different experiments that not clarify the proposed questions with several unclear points (13C-labeled experiments, FITR measurements, Carbon residues contribution, Photoluminescence lifetime experiments.)

So, I can't accept the paper in this way and I carefully recommend to the authors that read carefully the questions and answered it to no include more noise in this complex reaction.

1. Response: The lifetime was monitored at emission wavelength of 480 nm for comparison. Based on the photocatalytic performances, we believe that at least to some extent, the lifetime of the photo-generated electrons is long enough for charge transfer. Nevertheless, detailed study on the lifetimes of charge carriers for recombination, trapping, interfacial transfer, *etc*, is out of the scope of this manuscript. Besides, as we have mentioned in the manuscript, the S-scheme TiO₂/CsPbBr₃ heterostructure was formed in the hybridized semiconductors. The S-scheme enabled an internal electric field at interfaces, which drove electrons in TiO₂ CB and holes in CsPbBr₃ VB to combine, achieving more efficient charge separation and high redox capability of the remaining charge carriers.

Re-Response: I'm not in agreement with the author answer. First, the study of charge transfer carriers is not out of the scope when you want to demonstrate that has a S-type or other charge transfer scheme. Second, if you have performed the lifetime PL experiments at 480, you are only comparing the lifetimes of a TiO₂ luminescence signal. When QDs are deposited over TiO₂ a decreases in the luminescence lifetime is observed indicating a higher recombination rate. Therefore, this doesn't mean that you are...“achieving more efficient charge separation and high redox capability of the remaining charge carriers”. In order to demonstrate that “...an internal electric field at interfaces, which drove electrons in TiO₂ CB and holes in CsPbBr₃ VB...” you must include the lifetime at 520 nm (QD emission). If you observe an increase in the lifetime these could mean that TiO₂ is transferring electrons in TiO₂ CB and holes in CsPbBr₃ VB. So, please revise this part of the manuscript, repeat the experiments using a wavelength source distinctive for QDs and change your explanation because these data are not consistent with the conclusions.

Response: According to your suggestion, the luminescence lifetime was measured again, and the related explanation was also revised in the revised manuscript.

2. Response: (i) The FTIR spectra were measured with an attenuated total reflectance

(ATR) mode on Nicolet iS 50 (Thermo fisher, USA). Each spectrum was collected by merging 32 scans with a 4 cm^{-1} resolution. Calibration was conducted using a standard polystyrene sample prior to measurement. The IR measurement was repeated for 3 times on each sample, to exclude any possible technique error. Therefore, we believe that the IR shift of 4 cm^{-1} is reliable. (ii) The XPS analyses show that the amount of (Ti)-OH for TiO_2 decreased after hybridization with QDs, consistent with the FTIR result. Both XPS and FTIR results prove the chemical interaction between (Ti)-OH and CsPdBr_3 QDs, *i.e.*, the formation of Ti-O-Br linkage.

Re-Response. (i) Regarding to FTIR (ATR). First, -OH region of TiO_2 is very strange, hydrosil group in TiO_2 exhibit a sharp peak at higher wavenumber than water molecules (which should have a high intensity and broad), but in your case this peak is almost non-existent. The disappearance of this “peak” can be also attributed to the deposition of QD particles over the TiO_2 surface (but don't demonstrate the interaction). In addition, not important changes are observed on the wavenumber of -OH signal to conclude an interaction with QDs. The changes in the intensity of the broad band from 3000 to 3500 cm^{-1} are due to variations in the humidity of the sample. (ii) On the other hand, the changes in the XPS O1s signal are also unclear. Revising carefully the O1s signal of XPS experiments some assignments are not well defined. In the case of O1s peak of TiO_2 , authors ascribed a broad ca 528-534 eV to -OH group. This is not totally true. Even when this contribution is difficult to convolute, this is due to oxygenates (that also include to adsorbed water of other oxygenate organic compounds). So the decrease of this signal can be also attributed to an increase in of QD over the TiO_2 surface or even to a decrease in the adsorbed water (which is in agreement with the H_2O broad band decrease in FTIR). But to conclude that it is primarily due to the interaction of QDs with the OH groups of TiO_2 cannot be firmly affirmed. Revise both part of the manuscript, the XPS, FTIR assignment and of course the conclusion, because with this data is not possible to conclude a chemical interaction between QDs and TiO_2 .

Response: We completely agree your viewpoint that there is no chemical interaction between QDs and TiO_2 . We have updated the discussion and conclusion regarding FTIR and XPS results in the revised manuscript.

3. Response: (i) The photocatalytic CO_2 reduction was performed in a gas atmosphere in a homemade container, where oxygen was completely removed by high-purity N_2 (99.99%) flux prior to the experiment. The CO_2 gas was in situ produced through the reaction of NaHCO_3 powder and 2 M of H_2SO_4 , and the formed gas reagents diffused through a channel to the photocatalysts (Figure R1). Under this condition, the perovskite QDs were not in contact with liquid water and thus remained reasonably stable even after a few cycles.

(ii) The atmosphere in the system contains $\sim 3\%$ H_2O and $\sim 97\%$ N_2/CO_2 , considering that the equilibrium pressure of pure water at 25 °C is 3.169 kPa. The mixed N_2/CO_2 gas as the major component was nondestructive to CsPbBr_3 .

(iii) Previous work suggested that CsPbBr₃ can be stabilized by hydrophobic organic groups [Nat. Energy, 1 (2016), 15016.]. The CsPbBr₃ QDs here were terminated by the residual oleic acid species, as evidenced by FTIR results (Figure S5) and elemental analyses (Table R1).

Table R1. Elemental analysis of T, TC2 and C with an elemental analyzer.

	C	H	N
	(wt.%)	(wt.%)	(wt.%)
T	0.49	0.14	0.09
TC2	1.78	0.27	0.09
C	3.61	0.6	0.16

Re-response: (i) The only way to eliminate the all O₂ in a glass reactor is using vacuum pumps, performing cold/heat cycles (with N₂ liquid) and using vacuum joints. In addition, even in the case that you are using high purity gases (99.9999%) these N₂ coming from de air so you have O₂ in you cylinders (some ppms) which are highly important in the recombination reactions. Moreover, your H₂SO₄ solution has O₂ solved. So you cannot affirm that O₂ completely removed from the reactor, may be you cannot measured. In addition, it is still not explained with is the column that you are using to determine the O₂. To avoid uncertain, please include a comparison with the N₂/O₂ ratio. On the other hand, if your joints are not enough precise you could have a leak and this could be a possible explanation to not detect H₂. Other possibility to not detect H₂ is if you are using He as carrier gas in the GC. Even when major component is N₂ and CO₂, H₂O and O₂ or even H₂SO₄ vapor can destroy your sample.

Response: The CO₂ photoreduction activity of resultant samples was again measured in a closed gas circulation system (Supplementary Figure 7) with a Quartz and Pyrex glass hybrid reaction cell (Supplementary Figure 8). The related content can be found in the revised manuscript.

(ii) I partially agree with your answer. It is possible that these compound increase the hydrophobic character of CsPbBr₃ QDs but could have other negative effects in the photocatalytic performance. In addition, the C percentages measured by Elemental Analysis are too high taking into account your production yields. To avoid uncertain include in a balance of the C-based products (CH₄, CH₃OH and CO) if only come from these carbon residues, and compare with your results. The calculation offer in answer 6 is not finessed (revise my Re-response to this question).

Response: The FTIR spectra of TC2 before and after reaction were presented in Supplementary Figure 14. The characteristic absorbance bands of the aliphatic species from QDs showed no obvious variation, implying that the capping agent of QDs is stable and is not decomposed during the photocatalytic CO₂ reduction. GC-MS results also indicated the TC2 sample with good stability.

(iii) In addition to assure the answer (ii) and (iii) FTIR after reaction must be included to corroborate the presence of organic residues.

Response: The FTIR spectra of TC2 before and after reaction were presented in Supplementary Figure 14.

4. Response: The GC–MS analyses were carried out on a gas chromatography-mass spectrometer (6890N network GC system-5975 inert mass selective detector, Agilent Technologies, USA) equipped with an Agilent HP-5 capillary column (30.0 m × 0.32 mm i.d., film thickness 0.25 µm). Helium was used as the carrier gas. The column was maintained at 50 °C for 1 min and then heated to 100 °C at a programming rate of 8 °C min⁻¹. The temperature of injector was set to be 250 °C.

Re-Response: The Column (HP-5) is not adequate to separate permanent gases (such as H₂, CO, O₂, CH₄...). In addition the operation program including the starting temperature (50°C) is too high for gas separation. To avoid incertitude, please include the chromatogram of these experiments Furthermore, in the Figure S7b only CO₂ signals are shown. What's happened with products?

Response: The isotope labeling experiment was again carried out by gas chromatography-mass spectrometry (JMS-K9, JEOL-GCQMS, Japan and 6890N Network GC system, Agilent Technologies, USA) equipped with the column for detecting the products of ¹³CO (HP-MOLESIEVE, 30 m× 0.32 mm × 25 µm). The related content was added into the revised manuscript and Supporting information.

5. Response: Please refer to our response to your 3rd comment

Re-response: This answer is not completely response in 3rd comment. Please include FTIR spectra after reaction.

Response: The FTIR spectra of TC2 before and after reaction were presented in Supplementary Figure 14.

6. Response: We have calculated the amount of C1 products (CH₄, CO, CH₃OH, etc) from the organic residues, provided that all the carbon-containing residues were transformed, was only 0.03 mmol. On the other hand, we would like to highlight again that, control experiment was carried out in the absence of CO₂ gas source to study whether the photocatalytic products were derived from CO₂ or the organic residues. No product can be detected under this condition, confirming that all the products originated from CO₂ reduction. Furthermore, the ¹³CO₂ isotope tracer experiments also confirmed the C1 products were exclusively from the CO₂ gas.

Re-Response. Many articles have shown that even only in presence of TiO₂ and in absence of CO₂ different products are observed (CO, CH₄...) mainly due to the

presence of carbonates or organic residues in the catalyst surface. Which is the difference in your case?

Response: Control experiments (Supplementary Figure 10 and Table 2) showed that neither H₂ nor CO was detected in dark or in the absence of CO₂, suggesting that the light irradiation and input CO₂ were indispensable for the photocatalytic reaction. The FTIR spectra of TC2 before and after reaction indicated that the capping agent of QDs was stable and was not decomposed during the photocatalytic CO₂ reduction. GC-MS results also indicated the TC2 sample with good stability.

Regarding to the ¹³CO₂ experiments as I commented before some uncertain are still not solved. In the case of your calculations of carbon-containing residues. This must be compared with your photocatalytic results. Taking into account the measured %C in your catalyst (1.78 %wt) you have 0.03 mmols or C. However your results are in micromole/g, so this means that you if you use 0.002 g in the experiments the C-based products ca be 1483.34 micromoles per gram, which is higher than your accumulative production even if you consider the summation of all recycled experiments.

Response: We used MS signals to qualitatively analyze whether the products originate from the reduction of ¹³CO₂ or organic residues. The ¹³CO₂ GC-MS experimental results indicated that the input CO₂ was the carbon source of the products.

7. Response: Please refer to our response to your 1st comment.

Re-response. As I commented in 1st comment this question is not fully addressed. The decrease on Fluorescence lifetime means that the recombination rate is increased. You are focusing only in the TiO₂ signal (480 nm), but you don't know where go the electrons. If yo see an increase in the Florescence of QDs or in his florescence lifetime you could assign these changes to a possible charge transfer from TiO₂ to QDs.

Response: According to your suggestion, the luminescence lifetime was measured again, and the related explanation was also revised in the revised manuscript.